# Measurement report: Vertical distribution of atmospheric particulate matter within the urban boundary layer in southern China: Size-segregated chemical composition and secondary formation through cloud processing and heterogeneous reactions

Shengzhen Zhou[1,8*], Luolin Wu[1], Junchen Guo[1], Weihua Chen[2], Xuemei Wang[2*], Jun Zhao[1], Yafang Cheng[2,7], Zuzhao Huang[3], Jinpu Zhang[4], Yele Sun[5], Pingqing Fu[6], Shiguo Jia[1], Jun Tao[9], Yanning Chen[4], Junxia Kuang[4]

[1] School of Atmospheric Sciences, and Guangdong Province Key Laboratory for Climate Change and Natural Disaster Studies, Sun Yat-sen University, Guangzhou, 510275, P. R. China

[2] Institute for Environmental and Climate Research, Jinan University, Guangzhou, 511443, P. R. China

[3] Guangzhou Environmental Technology Center, Guangzhou, 510180, P. R. China

[4] Guangzhou Environmental Monitoring Center, Guangzhou, 510030, P. R. China

[5] State Key Laboratory of Atmospheric Boundary Layer Physics and Atmospheric Chemistry, Institute of Atmospheric Physics, Chinese Academy of Sciences, Beijing, 100029, P. R. China

[6] Institute of Surface-Earth System Science, Tianjin University, Tianjin, 300072, P. R. China

[7] Multiphase Chemistry Department, Max Planck Institute for Chemistry, P.O. Box 3060, Mainz, 55128, Germany.

[8] Southern Marine Science and Engineering Guangdong Laboratory (Zhuhai), 519082, P. R. China

[9] South China Institute of Environmental Sciences, Ministry of Ecology and Environment, Guangzhou, 510655, China

*Correspondence to*: Shengzhen Zhou (zhoushzh3@mail.sysu.edu.cn) and Xuemei Wang (eeswxm@mail.sysu.edu.cn)

**Abstract.** Many studies have recently been made on understanding the sources and formation mechanisms of atmospheric aerosols at ground level. However, vertical profiles and sources of size-resolved particulate matter within the urban boundary layer are still lacking. In this study, vertical distribution characteristics of size-segregated particles was investigated at three observation platforms (ground level, 118 m and 488 m) on the 610-meter-high Canton Tower in Guangzhou, China. Size-segregated aerosol samples were simultaneously collected at the three levels in autumn and winter. Major aerosol components, including water-soluble ions, organic carbon and elemental carbon, were measured. The results showed that daily average fine-particle concentrations generally decreased with

height. Concentrations of sulfate and ammonium in fine particles displayed small vertical gradients and nitrate concentrations increased with height in autumn, while the chemical components showed greater variations in winter than in autumn. The size distributions of sulfate and ammonium in both seasons were characterized by dominant unimodal mode with peaks at the size range of 0.44–1.0 μm. In autumn, the nitrate size distribution was bi-modal, peaking at 0.44–1.0 μm and 2.5–10 μm, while in winter it was unimodal, implying that the formation mechanisms for nitrate particles were different in the two seasons. Our results suggest that the majority of the sulfate and nitrate is formed from aqueous-phase reactions, and we attribute coarse mode nitrate formation at the measurement site to the heterogeneous reactions of gaseous nitric acid on existing sea-derived coarse particles in autumn. Case studies further showed that atmospheric aqueous-phase and heterogeneous reactions could be important mechanisms for sulfate and nitrate formation, which, in combination with adverse weather conditions, such as temperature inversion and calm wind, led to haze formation during autumn and winter in the Pearl River Delta (PRD) region.

## 1 Introduction

Air pollution is of serious environmental concern in China and is often characterized by high concentrations of many pollutants, among which fine particulate matter (particles with the aerodynamic diameter of 2.5 μm and smaller or $PM_{2.5}$) is currently the primary pollutant in most cities. Aerosol particles can profoundly affect public health, visibility, and climate change, and their effects are strongly dependent on size distribution and chemical composition (Pöschl, 2005; Zhang et al., 2015). The chemical constituents of particulate matter (PM) include sulfate, nitrate, ammonium, organic matter, elemental carbon, crustal species, and trace metals, which have a variety of primary and secondary sources in both nature and human activities. Moreover, primary and secondary aerosols undergo chemical and physical processes, for example, transport, cloud processing, and removal from the atmosphere, leading to significantly spatial and temporal variations of the sources and formation mechanisms of atmospheric aerosols (Huang et al., 2014; Sun et al., 2015; Zhang et al., 2015; Liang et al., 2016).

Knowing the size-resolved PM chemical composition is a key factor in understanding the sources, formation, and transformation of atmospheric particles (Cabada et al., 2004; Seinfeld and Pandis, 2006; Wang et al., 2014). Atmospheric aerosol number size distribution is characterized by a number of modes, such as the Aitken and nucleation modes (less than 0.1 μm), accumulation mode (~0.1 to ~2 μm), and coarse mode (~2 to ~50 μm). However, the mass distribution of airborne particles is typically dominated by two modes: the accumulation mode and the coarse mode (Seinfeld and Pandis, 2006). In many cases, accumulation mode consists of two overlapping sub-modes: condensation mode and droplet mode. The condensation submode originates from primary emissions and growth of smaller particles by coagulation and condensation, while droplet submode mainly results from cloud/fog processing or coagulation of smaller particles (Seinfeld and Pandis, 2006). The two sub-modes were first reported for sulfate particles ($0.2 \pm 0.02$ μm for the condensation mode and $0.54 \pm 0.07$ μm for the droplet mode) (Hering and Friedlander, 1982). Numerous studies have shown that in-cloud processes or multiphase reactions are plausible mechanisms for the formation of droplet-mode particles (Meng and Seinfeld, 1994; Zhuang et al., 1999a; Yao et al., 2003; Guo et al., 2010; Tian et al., 2016). Recently, strong evidences have been shown that the first sub-mode (i.e., condensation mode) can also be formed by cloud processing (Hoppel et al., 1985; Ovadnevaite et al., 2017). However, the contribution of smaller size particles in condensation mode to total aerosol mass concentration was found to be quite small. Coarse-mode particles are primarily produced by mechanical processes like sea spay, mineral particles and plant debris; however, coarse-mode secondary sulfates and nitrates have also been observed, and their formation has been attributed to heterogeneous and multiphase reactions (Pakkanen, 1996; Liu et al., 2008).

Measurements of ambient particles at several heights, rather than at a single ground level, provide unique information about their sources and dynamic transport. In addition, vertical PM distribution can reflect the influences of atmospheric boundary meteorology on aerosol chemistry. Vertical profiles of atmospheric pollutants are frequently measured in tall towers located in urban areas. Valiulis et al. (2002) estimated the trace metal emissions in Vilnius city using a vertical concentration gradient based on a TV tower and road tunnel measurement data and showed that traffic was the main source for airborne trace metals. Harrison et al. (2012) reported a wide measured range of ambient particle

physical properties and chemical compositions on the BT Tower 160 m above street level in central London. Oztürk et al. (2013) conducted high-resolution measurements of aerosol particle composition using a compact time-of-flight aerosol mass spectrometer and found considerable variability in the vertical distribution of aerosol mass concentration and composition on a 265-m tall tower near suburban Denver, Colorado. Chan et al. (2005) showed a complex vertical distribution of fine PM and carbonaceous species over the Beijing city based on measurements from a 325 m meteorological tower. Sun et al. (2015) conducted real-time and simultaneous vertical measurements of aerosol particles at ground level and at 260 m on the same tower, and showed very dynamic changes in vertical concentration profiles of meteorological parameters below 300 m that affected the formation and evolution processes of aerosols during haze episodes (Sun et al., 2015; Wang et al., 2018). A series of vertical measurements of atmospheric particulate matter and meteorological parameters on a 255 m meteorological tower in the Tianjin have been carried out in recent years (Zhang et al., 2011; Shi et al., 2012; Wang et al., 2016). Deng et al. (2014) reported the vertical distribution of $PM_{10}$, $PM_{2.5}$ and $PM_{1.0}$ mass concentrations measured on Canton Tower at 121 m and 454 m from November 2010 to May 2013. However, measurements of size-resolved chemical composition in the vertical within the urban boundary layer are still lacking. Wang et al. (2016) investigated the size distribution of chemical compositions and sources of particulate matter in different modes at ground level and 220 m in Tianjin. They suggested that 220 m is insufficiently high to eliminate the influence of local surface emissions and measurements taken at that height do not reflect the background levels of pollutants within urban canopy. Aerosol pollution frequently occurs in China, as exemplified by three cities groups in the Jing-Jin-Ji (Beijing, Tianjin, and Hebei province), the Yangtze River Delta, and the Pearl River Delta regions. State–of–the–art air quality models still often fail to simulate the observed high $PM_{2.5}$ concentrations even after including aerosol-radiation-meteorology feedback, indicating that key atmospheric chemical processes, such as heterogeneous and multiphase reactions, are lacking in models for secondary aerosol formation (Zheng et al., 2015; Cheng et al., 2016). To improve the understanding of haze formation, models will require updated kinetic and mechanistic data of multiphase chemistry, and quantification of the aerosol formation through heterogeneous reactions are needed from field measurements, laboratory experiments and model simulations under real atmospheric conditions (Zheng et al., 2015; Yun et al.,

2018; An et al., 2019). Field studies showed that extremely high $PM_{2.5}$ concentrations usually occurred under high relative humidity conditions (Sun et al., 2014; Wang et al., 2014). Multiphase reactions in the cloud liquid water and in aerosol water can promote secondary aerosol formation (Seinfeld et al., 2006; Ervens, 2015; McNeill, 2015; Cheng et al., 2016). Hence, size-resolved vertical concentration profiles of major components of PM can add to important knowledge of aerosol sources and formation mechanisms in the urban atmospheric boundary layer.

The Pearl River Delta (PRD) region is the low-lying area surrounding the Pearl River Estuary, where the Pearl River flows into the South China Sea. The weather is generally warm and humid all year and is strongly influenced by the Asian monsoon. The PRD region is one of the most densely urbanized regions in the world, and has recently experienced severe PM pollution and photochemical smog events (Zhang et al., 2008; Huang et al., 2014; Zhou et al., 2014). Autumn and winter are typical pollution seasons in this region (Chan et al., 2008). In this study, size-resolved PM samples were collected at three heights (ground level, 118 m, and 488 m) on Canton Tower in Guangzhou, the central city in the PRD region. The main water-soluble ions ($Na^+$, $K^+$, $Mg^{2+}$, $Ca^{2+}$, $NH_4^+$, $F^-$, $Cl^-$, $NO_2^-$, $NO_3^-$, $SO_4^{2-}$) and carbonaceous species (organic carbon, OC; elemental carbon, EC) were measured, and these observations are used to examine formation mechanisms and sources. The objectives of this study are to (1) analyze the vertical mass size distribution of the PM chemical components and the factors that affect their vertical variations; and (2) investigate the roles of in-cloud processes and multiphase reactions in secondary aerosol formation and the implication for haze pollution in subtropical urban areas. (3) Evaluating the simulation performance of WRF-Chem model in the vertical based on the measurement data.

**2 Methodology**

**2.1 Observational site and sample collection**

The sampling site, Canton Tower, is located in central urban Guangzhou and is the second highest TV tower in the world with a total height of 610 m. The main tower is 454 m high and the antenna mast adds another 156 m. Four levels (ground level, 118 m, 168 m, and 488 m) were selected by the Guangzhou Environmental Monitoring Center (EMC) to create a vertical gradient of observation platform. Online measurements of pollutants including $SO_2$ (model 43i, Thermo), CO (model 48i,

Thermo), $O_3$ (model 49i, Thermo), $NO/NO_x$ (model 42i, Thermo), PM ($PM_{2.5}$ and $PM_{1.0}$, SHARP 5030, Thermo) were conducted on this four-layer observation platform. Meteorological factors (relative humidity, temperature, wind speed and direction) were also recorded on the tower. All these data were applied for the discussions in the following sections.

5   Size-segregated aerosol samples were concurrently collected at three of the four levels (i.e., ground level, 118 m and 488 m) in autumn (October and November 2015) and winter (December 2015 and January 2016) (Fig. 1). Three six-stage samplers (Model 131 High-Flow Impactor, MSP Corporation) with a sampling flow rate of 100 L min$^{-1}$ were used at the three heights. The 50% cut-point diameters of the six-stage sampler were 0.25, 0.44, 1.0, 1.4, 2.5, 10.0, and 18.0 (inlet) μm. The three impactors (or

10 samplers) were calibrated using mass flow meter (TSI, model 4040) in laboratory prior to the measurements. The flow rates of the impactors were measured at the beginning of the sampling. At the end of the sampling, the flow rates were recorded again. If the flow rate of each impactor at the beginning and end of the sampling differed by more than 10%, the sample was marked as suspect and the data was discarded. The average flow rates at the beginning and end of the sampling was used to be

15 the sampling flow rate. In addition, a magnehelic pressure gauge was used to monitor the inlet flow rate through the impactor. The pressure drop was also recorded at the beginning and end of the sampling. A 24-h sampling resolution was adopted every other day from 10:00 a.m. to 10:00 a.m. the next day (local time, UTC + 8). The collection substrates were 75-mm diameter quartz filters and a final 90-mm quartz filter was applied to collect aerosols with diameters of less than 0.25 μm. To eliminate possible organic

20 contaminants, all of the quartz membrane filters were prebaked at 550 °C for 4 hours before use. In total, 19 and 13 sets of samples, including one set of background samples, were collected at each height in autumn and winter. After collection, the filters were put into Petri dishes, kept in ice boxes during transportation to the laboratory and then stored in a refrigerator at -18 °C prior to analysis. We estimated the impacts of temperature and pressure on the flow rate due to the sampling heights which

25 are less than 5% (Refer to supplementary).

**2.2 Chemical analysis**

   A quarter of each quartz filter was cut out and dissolved in 15 mL of deionized water (18.2 MΩ, Millipore) for 30 min in an ultrasonic ice water bath. The extracted solution was filtered through a

microporous membrane (pore size, 0.2 μm) into a clean polycarbonate bottle, and then analyzed by an ion-chromatograph (ICS-5000, Dionex). The cations ($Na^+$, $K^+$, $Ca^{2+}$, $Mg^{2+}$, $NH_4^+$) were separated using a CS12A column (4 × 250 mm) and eluted with KOH solution. The anions ($F^-$, $Cl^-$, $NO_3^-$, $NO_2^-$ and $SO_4^{2-}$) were analyzed using an AS23 column (4 × 250 mm) and eluted with a methane sulfonic acid solution. Multiple points of calibration were used for each batch of ionic analysis. The OC and EC mass concentrations were determined using a thermal optical carbon analyzer (DRI Model 2001A, Atmospheric Inc., USA). The analytical procedures were described in detail in Chow et al. (2001) and Cao et al. (2004). Due to the non-uniform deposition nature of the size-resolved samplers, charring correction using optical transmittance may introduce uncertainty in determining the OC and EC split point (Huang et al., 2009). The data presented in this paper were all field-blank corrected.

## 2.3 Data analysis

Back-trajectory analysis was performed using the Hybrid Single-Particle Lagrangian Integrated Trajectory Model (HYSPLIT 4.9). The principle of this model can be found in Draxler and Hess (1998). The Global Data Assimilation System (GDAS, 1°×1°) was used for the input meteorological data, and 72-h air mass back-trajectories were calculated at starting times of 02:00, 08:00, 14:00, and 20:00 (UTC), with arrival heights of 200 and 500 m above ground level. Cluster analysis was performed to segregate the calculated trajectories into distinct cluster groups using the HYSPLIT clustering algorithm. Ceilometer (Model CL-31, Vaisala Corp, Finland) was applied to measure the vertical backscatter density on the roof of South China Institute of Environment Sciences, Ministry of Ecology and Environment, which is about 4 km northeast of the Canton tower.

Vertical profiles of wind direction and speed, relative humidity (RH), temperature (T) and chemical components of $PM_{2.5}$ were simulated by the Weather Research and Forecasting Model coupled with online chemistry (WRF-Chem) in version 3.7.1 (Skamarock et al., 2008). Detailed information on the model setup can be found in the supplementary materials and references (Fan et al., 2015; Chen et al., 2016).

## 3 Results and discussion

## 3.1 General characteristics

Figure 2 shows the temporal profiles of daily averaged $PM_{2.5}$ mass concentrations measured at the three heights (i.e., ground level, 118 m and 448 m) during the sampling periods. The mismatch between the real-time $PM_{2.5}$ concentrations and the reconstructed $PM_{2.5}$ mass by combining the main components was likely due to sampling artefacts and lack of comprehensive offline analysis of $PM_{2.5}$ chemical components such as H and O associated with OC, geological minerals, and liquid water (Chow et al., 2015). The daily averaged $PM_{2.5}$ mass concentrations on the three heights varied significantly in the ranges of 12.5–76.0 µg m$^{-3}$, 12.3–54.2 µg m$^{-3}$, and 7.9–44.4 µg m$^{-3}$ in autumn, and in the ranges of 10.2–104.8 µg m$^{-3}$, 10.7–83.4 µg m$^{-3}$, and 7.2–47.2 µg m$^{-3}$ in winter. The average $PM_{2.5}$ mass concentrations were 44 ± 15, 36 ± 11, and 28 ± 10 µg m$^{-3}$ at ground level, 118 m and 488 m in autumn, slightly higher than those in winter (42 ± 23, 34± 19, and 22 ± 12 µg m$^{-3}$). A pollution episode (i.e., E1) in autumn was identified when the $PM_{2.5}$ concentration at ground level exceeded the air quality standard (75 µg m$^{-3}$), and another episode (i.e., E2) was identified in winter when the standard was exceeded continuously over three day period. The diurnal variations of $PM_{2.5}$ and $PM_{1.0}$ concentrations at the three heights in autumn and winter are shown in Figs. S1 and S2. In general, $PM_{2.5}$ and $PM_{1.0}$ concentrations at ground level and 118 m showed distinct diurnal cycles with higher concentrations occurring at rush hours in the morning and evening. However, the concentrations at 488 m showed unimodal distribution with higher concentrations observed in the afternoon (12:00-17:00), lagging 3-4 hours behind those at ground level and 118 m. This can be attributed to the fact that the convective boundary layer begins to extend vertically after sunrise on typical days, and particles were transported upward by turbulence. The diurnal variations of CO and $NO_x$ showed similar trends as the PM. The $O_3$ diurnal cycle showed a single peak pattern at the three levels with the highest values at around 14:00 LST. The $O_3$ concentrations were higher at 488 m than at the lower levels. The $O_3$ concentration differences between the lower levels and 448 m were widened at night due to the intensive NO titration loss at lower levels (Figs. S1 and S2).

**3.2 Vertical distribution**

**3.2.1 Vertical distribution of the major chemical components**

The profiles of the major PM$_{2.5}$ chemical components can generally be classified into three vertical gradients. The first category presents the highest concentration at ground level. The second category shows the highest concentrations at 118 m. And, the third category shows the highest concentration at 488 m. The statistics of the three types in autumn and winter are listed in Table S1 and S2. We found that the second and third categories were the major categories for sulfate, nitrate and ammonium (SNA) in autumn, while those in winter were prone to peak at the ground and 118 m. Meanwhile, the OC and EC presented the highest concentration at ground level in both seasons.

Figure 3 shows the representative and average vertical profiles of PM$_{2.5}$, sulfate, ammonium, nitrate, OC, and EC mass concentration at the tower. In autumn, the vertical gradients for averaged sulfate, nitrate and ammonium were observed to be shallow, attributed to the second category in which sulfate and nitrate concentrations were slightly higher at 118 m (Fig. 3a) while mean ammonium concentrations increased with height. Sulfate, nitrate, and ammonium concentrations on the polluted day (i.e., November 18, 2015) all increased with height. In particular, nitrate concentration was 1.5 times higher at 488 m than that at ground level, which will be further discussed in case studies. The vertical gradients for OC and EC in autumn were found to be different from those of sulfate and ammonium, with the EC concentration 27.9% lower and the OC concentration 34.0% lower at 488 m than at ground level respectively. The results suggest that the carbonaceous components are likely from local sources and the inorganics (sulfate and nitrate) are transported from some distances. The decrease in air pollutant concentrations with height is considered to be associated with ground-level sources (Zauli Sajani et al., 2018). No vertical gradients could be established for any of the measured PM components during cleaner days (e.g., as seen for October 31, 2015), which was likely attributed to the turbulent mixing of air pollutants within the boundary layer (Guinot et al., 2006).

In winter, averaged concentrations of sulfate and ammonium were generally observed to be higher at ground level than those at higher levels (Fig. 3b). However, concentrations of nitrate, OC and EC were peaking at 118 m. On cleaner days (i.e., Jan. 17, 2016) the vertical gradients for mean PM$_{2.5}$, SNA, OC, and EC mass concentrations were found to be shallow due probably to the well mixed air masses, while on polluted days (i.e., Jan. 2, 2016), the concentrations for sulfate, nitrate, ammonium and OC were higher at 118 m. Our results showed that the vertical gradients for sulfate, nitrate and ammonium

concentrations tend to peak at 118 m and 488 m in both autumn and winter seasons when the $PM_{2.5}$ concentrations were high (Table S1). The reasons are unclear, but they were probably due to local chemical formation or regional transport of particles. However, back trajectory analysis showed that local chemical formation contributed to high SNA mass concentrations rather than regional transport during the sampling time (Fig. S4).

### 3.2.2 Chemical composition in fine and coarse aerosols

Figure 4 shows the percentages of measured chemical composition in fine ($PM_{2.5}$) and coarse ($PM_{2.5-18}$) particles at the three heights (ground level, 118 m, and 488 m). Sulfate, OC, and EC were the major chemical components of fine particles in autumn. Elevated proportions of nitrate and ammonium were found in winter, possibly related to the equilibrium between gas phase $HNO_3$ and $NH_3$ and the particle phase. Ammonium nitrate is a temperature- and relative humidity-dependent compound, and low temperature and high RH facilitate the gas-to-particle partitioning (Wang et al., 2012; Bian et al., 2014). During the sampling periods, the average temperature in winter (13.5 °C) was much lower than that in autumn (23.1 °C) while the RH was vice versa, explaining higher concentration of $NH_4NO_3$ in winter than in autumn. There was no significant difference between the relative contributions of the main chemical components of fine particles at ground level and at 118 m. We found that the total contribution of $SO_4^{2-}$, $NO_3^-$ and $NH_4^+$ at 488 m was higher than the corresponding contributions at the two lower levels, indicating more favorable secondary formation and aging processes of aerosols at higher altitude. Our results showed that OC, nitrate, crustal (e.g., $Ca^{2+}$, $Mg^{2+}$) and sea salt (e.g., $Na^+$ and $Cl^-$) were the major components of the measured species, but the comparisons with $PM_1$ and $PM_{2.5}$ suggest these components were relatively small in the coarse particles. The percentages for nitrate and sea salts were higher in autumn than in winter, suggesting that sea salt is a nonnegligible source of aerosols in autumn in the PRD region. We also found that the fractions of primary inorganic ions (e.g., $Ca^{2+}$) and EC in coarse particles decreased with height, probably due to their sources (e.g., road dust and traffic emissions) being near the ground.

### 3.2.3 Mass size distributions

Difference in mass size distributions may be attributed to difference in the sources and formation mechanisms of aerosol chemical components. The average mass size distribution of the ionic compounds, OC and EC at the three heights during autumn and winter are shown in Figs. 5 and 6.

(1) Sulfate, nitrate, and ammonium

Sulfate did not show obvious seasonal and vertical variations in mass size distribution (Figs. 5 and 6). The average mass size distributions of $SO_4^{2-}$ showed a dominant peak in the range of 0.44–1.0 μm (a typical droplet mode) and a minor coarse mode in the range of 2.5–10 μm. The mass size distributions of sulfate at the three levels were similar, indicating that sulfate may have similar formation mechanisms at these levels. Previous studies showed that droplet mode sulfate could be formed in cloud or fog (Zhuang et al., 1999a; Kerminen and Wexler, 1995; Meng and Seinfeld, 1994). Guo et al. (2010) proposed three possible formation processes for droplet mode sulfate, including condensation and coagulation of smaller particles, in-cloud aqueous processes, and processes in deliquesced aerosol particles. It is generally recognized that coagulation is negligible at typical ambient particle number concentrations (Hinds, 1999). Therefore, in-cloud coalescence of droplets or aqueous phase chemistry is responsible for production of several sulfate modes (Feingold et al., 1996). Using a positive matrix factorization (PMF) method similar to that described in Guo et al. (2010), we obtained three modes (i.e., condensation, droplet, and coarse modes) for the sulfate size distribution. The droplet mode accounted for 79.4%, 78.5%, and 86.9% in autumn, and 78.5%, 78.3%, and 80.4% in winter at ground level, 118 m, and 488 m (Table S3). High relative humidity was measured during the autumn and winter measurement periods (at ~78% and 80% on average) in Guangzhou. The contribution of droplet-mode sulfate was higher at 488 m than that at the two lower levels, suggesting that in-cloud or aerosol droplet processes are likely to be the main formation pathways for sulfate. There is also evidence of frequent cloud coverages at 500-1500 m above the ground in urban Guangzhou measured using a ceilometer during the measurement periods (Figures S9 and S10). Relative humidity would influence the relative size distributions of the different chemical species. The air was not dried upstream of the impactor and therefore relative humidity would influence the size distributions of different chemical species. In this study, the monthly averaged relative humidity was around 80% in both seasons. We assume that the hygroscopic growth factor is independent of particle size and is about 1.5 for ammonium sulfate and

ammonium nitrate at 80% RH (Tang, 1996). A maximum overestimated value of about 10%, 13% and 6% for droplet mode sulfate was obtained at ground level, 118 m and 488 m, respectively. The droplet mode nitrates were overestimated to be about 11%, 16%, and 6% at ground level, 118 m and 488 m, respectively. Our results are similar to those in Chen el al (2018), which concluded that the influence of cut-off shift on filter-based particle sampling driven by hygroscopic growth is generally negligible (less than 7%) in urban areas, but need to be considered (about 10–20%) in continental background areas in Europe. The influence from relative humidity on the size distribution is indeed present based on the average particle concentrations in the droplet mode in this study. However, it is unlikely to change our conclusion that in-cloud processing contributed to the droplet mode aerosols (Table S3). In addition, Meng and Seinfeld (1994) showed that water accretion alone cannot account for the growth of droplet-mode particles from the condensation mode. They proposed that activation of condensation mode particles to form fogs or clouds followed by significant production of sulfate from the $SO_2$ oxidation within droplets (also for nitrate) and fog evaporation are plausible mechanisms for the formation of urban and regional aerosols in the droplet mode.

The mass size distribution for nitrate exhibited two modes with mass median aerodynamic diameters at 0.44–1.0 μm (fine mode) and 2.5–10 μm (coarse mode) in autumn, while the fine mode pattern peak, at around 1.0 μm, was observed in winter. The coarse-mode nitrate was likely formed by the heterogeneous reactions of gaseous nitric acid with pre-existing sea- and soil-derived coarse particles (Anlauf et al., 2006; Harrison and Pio, 1983; Harrison and Kitto, 1990; Pakkanen, 1996; Wall et al., 1988; Wu and Okada, 1994; Zhuang et al., 1999a). We found that coarse-mode $Na^+$, $Cl^-$, and $NO_3^-$ were at almost the same particle size, while $Ca^{2+}$ peaked at a particle size larger than $NO_3^-$ (Fig. S3). It is thus reasonable to conclude that coarse-mode $NO_3^-$ is probably associated with sea salt rather than $Ca^{2+}$, which is consistent with the previous work in Hong Kong (Zhuang et al., 1999b). Sea salt particles can grow by water uptake in fogs and clouds. A previous study showed that a substantial amount of nitrates forms when $HNO_3$ reacts with deliquesced sea-salt as compared to the dry NaCl particles (Brink, 1998). Hence, we speculated that nitrates were formed from the reactive uptake of $HNO_3$ in the deliquesced sea salt droplets rather than dry particles in Guangzhou. The back-trajectory cluster analysis showed that the sampled air masses were predominantly from the South China Sea and moved toward

Guangzhou in autumn (Fig. S4), bringing high concentrations of sea salt particles available for heterogeneous reactions. Moreover, high relative humidity, fog, and low clouds which were observed during the observation, could facilitate the heterogeneous formation of coarse-mode nitrates.

In winter, the mass size distribution for nitrate was dominated by the droplet mode (with a size range of 0.44–1.0 μm), in contrast to a relatively small nitrate peak in this mode in autumn (Figs. 5 and 6). Previous studies showed that droplet-mode nitrate could be produced by the condensation of nitric acid onto pre-existing particles and/or heterogeneous reactions of $N_2O_5$ (Guo et al., 2010; Wang et al., 2012). The contribution of heterogeneous reactions between nitric acid and sea salt droplets was minor in winter given that the air masses were predominantly from inland (Fig. S4), bringing a much lower concentration of coarse- mode sea salt aerosols than that in autumn (Figs. 5 and 6). Therefore, the heterogeneous reactions of nitric acid with sea salt droplets were less prominent in winter.

Ammonium commonly appears in the forms of $NH_4NO_3$ and $(NH_4)_2SO_4$. Although fine-mode $NH_4^+$ ions were found to be dominant in the mass size distribution in both autumn and winter, the concentration of fine-mode $NH_4^+$ ions was strongly correlated with that of the same mode $SO_4^{2-}$ (R > 0.89 at three heights), while only moderately correlated with that of fine-mode $NO_3^-$ ions in autumn, implying that fine-mode nitrates exist not only as $NH_4NO_3$ but also as other forms (i.e., $NaNO_3$) in autumn. Figure 5 shows ammonium concentration was the highest in 488 m in autumn. The possible reason for this phenomenon might be that temperature (T) was lower and relative humidity (RH) was higher at 488 m, which was favorable for the partitioning of semi-volatile $NH_4NO_3$ into particle phase (Stelson and Seinfeld, 1982; Wang et al., 2012). This is supported by the evidence that nitrate concentration in fine particles generally increased with height (Figure 3).

(2) Sodium and chloride

The mass size distributions for $Na^+$ and $Cl^-$ showed distinctly different patterns in autumn and in winter. $Na^+$ and $Cl^-$ exhibited unimodal peaking in the range of 2.5–10 μm in autumn. The proportion of $Na^+$ in coarse mode of the total $Na^+$ mass was 63%, 71%, and 68%, and the proportion $Cl^-$ ions observed in coarse mode was 59%, 69%, and 70% respective to ground level, 118 m, and 488 m. This indicates that their contributions at the upper levels were slightly higher than those at ground level. Chloride in the coarse-mode particles was thought to originate from marine sources and be associated with sodium

(Zhao and Gao, 2008; Bian et al., 2014). As discussed above (Fig. S4), ambient air in Guangzhou was strongly influenced by particles from marine sources which were transported from the South China Sea in autumn. We calculated the chloride depletion based on the concentrations of $Na^+$ and $Cl^-$ (Eq. 2) to estimate the nitrate and sulfate formation rates on the sea-salt particles (Zhuang et al., 1999b):

$$[Cl_{dep}] = (1.174[Na^+] - [Cl^-])/1.174[Na^+] \times 100\% \qquad (2)$$

where $[Na^+]$ is the measured concentration and 1.174 is the molar ratio of $Cl^-$ to $Na^+$ in seawater. Higher chloride depletion (in percentage) was found for larger size of particles (> 1.0 μm) at the highest testing level (488m) (Fig. S5), demonstrating more aged aerosol which had undergone significant chemical processing during advection to the 488 m sampling site. In comparison, possible sources for fine-mode chloride particles were biomass burning, coal combustion and waste incineration (Fu et al., 2018; Zhao and Gao, 2008). The origin of fine chloride particles in winter during the measurement periods, however, is difficult to determine based on the broad peaks for $Na^+$ and $Cl^-$.

(3) OC and EC

In general, the mass size distributions for OC showed dominant droplet modes in both autumn and winter, with smaller peaks in coarse mode (Figs. 5 and 6). The size distributions of OC in our work are similar to those in previous studies in Shanghai (Ding et al., 2017) and at Alpine valleys (Jaffrezo et al, 2005), while they are very different from those at rural and remote sites at Tibetan Plateau (Wan et al., 2015) and Xishuangbanna in China (Guo, 2016). The dominant droplet mode for OC is very similar to that of sulfate, suggesting that a large amount of OC may originate from secondary formation processes (Jaffrezo et al., 2005). The high OC/EC ratios were found in particles with sizes larger than 0.25 μm, especially for droplet mode particles, indicating the enhancement of the secondary formation of OC in this mode. We further evaluated the contributions of secondary organic carbon (SOC) to OC using EC-tracer method (Castro et al., 1999; Zhou et al. 2014). The results showed that SOC accounted for a large fraction of OC in our study (Figure S6). The mass size distributions for EC are fairly different between autumn and winter. Huang et al (2006) reported that EC showed a dominant accumulation mode with a mass median aerodynamic diameter of 0.42 μm from vehicle emissions in a tunnel in Guangzhou. However, in this study, the peak concentration for EC in autumn was in the size range of 0.44-1.0 μm,

suggesting that EC has been aged after emission. EC showed a general bimodal peak in winter, with a broad EC peak in accumulation mode and a sharp peak (the highest concentration) at around 1.0–1.44 μm at 188 m, suggesting that its source might be different from the other two levels. One possible reason for the abnormally high concentrations of EC was the influence of local point sources (i.e., high chimneys from power plants and industry) around Guangzhou which emitted elevated concentrations of air pollutants from combustion sources. We also found that the concentrations of co-emitted $SO_2$ and CO were the highest at 188 m among the three heights (Fig. S7).

### 3.3 Case studies of PM vertical profile

Factors that influence vertical distribution of PM include meteorology, regional transport, source emissions, and chemical reactions. In the previous sections, our results revealed that in-cloud aqueous reactions and heterogeneous reactions were important aerosol formation pathways in this coastal urban area under high relative humidity. Here we consider two PM pollution episodes (E1 and E2) to investigate these mechanisms for haze formation in autumn and in winter (Fig. 2). There was no rain during the pollution episodes. The E1 episode which occurred on November 18, 2015 represented a typical pollution scenario in autumn. An anomalous increase in $PM_{2.5}$ concentration was observed at 168 m, as compared to the concentrations on non-event days. In addition, the average sulfate, nitrate, and ammonium (SNA) concentrations were higher at 488 m and 118 m than those at ground level (Fig. 3a). We simulated horizontal and vertical wind, RH, and T. The results showed that horizontal southerly wind was prevalent prior to November 18 with a period of calm wind from around 2:00 LST to 14:00 LST of the day. Subsequently, the wind direction changed predominantly to northerly (Fig. S8a), consistent with back trajectory analysis which showed that air masses firstly came from the south and then changed direction to the north on November 19 (Fig. S9). Low altitude temperature inversion was observed between 118 m and 168 m that night probably due to the convergence of two different air streams (Fig. S8a). A previous study (Wu et al., 2015) demonstrated that poor air quality is associated with surface and low-altitude inversions in the PRD region. The RH vertical profile decreased from ground level to 168 m and became relatively stable between 168 m and 488 m. Subsequently the RH increased until it reached maximum at around 900 m, followed by a sharp decrease (Fig. 7a). We simulated a large amount of low cloud cover on November 17 and 18 based on WRF model results and

MODIS satellite images (Fig. 8a and Fig. S10). The vertical wind blew dominantly upward during nighttime and downward during daytime (Fig. 8a), which facilitated transport of residual particles produced from cloud evaporation to lower altitudes after sunrise. The average sulfur oxidation ratio (SOR = $n\text{-}SO_4^{2-}$ / ($n\text{-}SO_4^{2-}$ + $n\text{-}SO_2$) during E1 was 0.22, 0.18, and 0.12 at ground level, 118 m, and 488 m, higher than that on non-event days. The concentration of $SO_2$ increased with height (12.4 µg m$^{-3}$ at ground level, 16.1 µg m$^{-3}$ at 118 m and 27.0 µg m$^{-3}$ at 488 m), suggesting that it was impacted more by emissions from local sources, where the air masses were fresh. The newly emitted $SO_2$ at high altitudes compensated for part of the sulfate converted from $SO_2$, leading to decrease of SOR with heights. The corresponding values for nitrogen oxidation ration (NOR = $n\text{-}NO_3^-$ / ($n\text{-}NO_3^-$ + $n\text{-}NO_2$)) were 0.01, 0.02, and 0.07 at the three levels. A number of previous studies demonstrated that high relative humidity favors the production of secondary aerosols (Sun et al., 2013; Cheng et al., 2016; Zheng et al., 2016). Our results suggest that aqueous-phase and heterogeneous reactions contributed significantly to the sulfate and nitrate in the PRD region during this episode.

The E2 episode represented a typical pollution event in winter, showing a similar PM$_{2.5}$ vertical distribution as E1. The highest sulfate, nitrate, ammonium, and OC concentrations were observed at 118 m (Fig. 3b). Horizontal wind was mainly from the north before noon (January 2) and changed to south in the afternoon (Fig. S8b). Similar temperature and RH profiles were found for E2 to those for E1 which was characterized by low altitude temperature inversion extending from 118 m to 488 m at January 2 and from 50 m to 168 m on January 3, as well as a higher RH at higher levels during the nighttime (Fig. 7b). Low-level cloud was observed during this period, associated with weak convection simulated by WRF (Fig. 8b and Fig. S11). The average SOR in PM$_{2.5}$ was 0.36, 0.27, and 0.30 at ground level, 118 m, and 488 m, again higher than on non-event days. The average NOR was 0.13, 0.14, and 0.21 at ground level, 118 m, and 488 m, twice or three times higher than on non-event days. Based on the above findings, a schematic graph was generated to illustrate one of the typical haze formation mechanisms in the PRD region in autumn and winter (Fig. 9). A calm wind zone was established over the PRD region during the later autumn and winter pollution episodes due to the confrontation of southerly and northerly air masses, which have potential to further transform into strong nocturnal temperature inversions. The stagnant atmospheric conditions inhibited the air pollution dispersion. Low-

level cloud cover facilitated the surface aerosol pollution due to in-cloud processing, where secondary aerosols were produced from the intensive aqueous reactions within the clouds and cloud evaporation and redistributed residual aerosols. Previous studies have shown high mixing ratios of gas phase hydroxyl (OH) and peroxy ($HO_2$, $RO_2$) radicals in the PRD region (Hofzumahaus et al., 2009; Lu et al., 2012). High concentrations of hydrogen peroxide ($H_2O_2$) and $O_3$ were also detected in this region (Hua et al., 2008; Wang et al., 2017). We did not measure these oxidants in either gas or aqueous phases. However, it is reasonable to assume that these gas phase oxidants might be scavenged by the clouds which are then transferred into the cloud droplets and facilitated the aqueous phase reactions. In addition, the temperature inversion layer disappeared during daytime and downward vertical wind speed was found through the model results (Fig. 8), leading to be under favorable meteorological conditions which facilitate the release and downward transport of residual aerosols from evaporating low-level clouds. The aforementioned processes were confirmed in our study which shows that the vertical concentrations of sulfate, nitrate, and ammonium increased with height during pollution episodes. Our results suggested that meteorology (such as nighttime temperature inversion and calm wind) together with aqueous phase (cloud processing) and heterogeneous reactions would significantly contribute to the aerosol formation and haze episodes in autumn and winter seasons over the PRD region.

## 3.4 Model simulation and implications

Vertical concentration distributions of sulfate, nitrate and ammonium were further simulated by WRF-Chem model. The description and configurations of the model can be found in the Supplementary. Figure 10 shows the simulated vertical concentration profiles of sulfate, nitrate and ammonium in autumn and winter and their comparisons with observation. Sulfate was generally underestimated in WRF-Chem model at the upper level, while it was in relatively good agreement with observation at the surface. Possible reasons for the underestimations of sulfate are: (1) $SO_2$ precursors were underestimated at the upper levels (by about 45% to 77%, table S6), possibly due to the insufficient upward transport of $SO_2$ in the current model, especially in urban area where the urban canopy is low in resolution; (2) heterogeneous/multiphase formations of sulfate in droplets or aerosol water have not been considered enough in current model (Chen et al., 2016; Cheng et al. 2016). Nitrate was overestimated by WRF-Chem model. Here three reasons were put forward: (1) the underestimation of

$SO_4^{2-}$ at the upper levels, which consumes less $NH_3$, facilitates the formation of $NH_3NO_4$ formation in the fine-mode (Tuccella et al., 2012); (2) heterogeneous reaction of $HNO_3$ on coarse-mode sea salt aerosols, however, will reduce the formation of fine-mode nitrate (Chen et al, 2016). Therefore, sea salt emissions in current model should be evaluated especially over the coastal regions; (3) the cloud fraction and liquid water content may not be well simulated in the model. For ammonium, the simulated concentrations were overall consistent with the measured ones except for being slightly overestimated at ground level. The large discrepancies between observation and simulation on sulfate and nitrate suggested that physical and chemical mechanisms in current WRF-Chem model still need to be improved to better predict aerosol mass and composition. Based on our observation, in-cloud aqueous phase reactions and heterogeneous reactions should play important roles in sulfate and nitrate formation, which need to be refined in the model. Evaluation of WRF-Chem model incorporating the above-mentioned mechanisms is beyond the scope of this study and in-depth investigation needs to be done in future. Hence, more studies, such as long-term aerosols and high frequency micrometeorological measurements (Valiulis et al., 2002; Ceburnis et al., 2008; Ervens, 2015), are needed to identify the key aerosol sources and formation pathways, and to further improve the air quality models.

## 4 Conclusions

Vertical characteristics and potential formation processes of size-resolved aerosols were studied during autumn and winter seasons utilizing the 610 m Canton Tower in Guangzhou. Complex vertical variations in PM composition were observed. In autumn, sulfate and ammonium had shallower vertical gradients than nitrate, which showed higher concentrations at higher observation levels. OC and EC showed steeper vertical gradients, with concentrations 34.0% and 27.9% lower, at 488 m than at ground level. The chemical components of the fine particles showed more pronounced and complex vertical gradients in winter than in autumn, possibly due to the effects of atmospheric stability, regional transport, and chemical reactions. The percentage of secondary inorganic ions in fine particles generally increased with height. The size distributions of sulfate and ammonium were similar at the three heights during the observation, characterized by a dominant droplet mode. Bi-modal size distributions in autumn and a unimodal mode in winter were observed for nitrate, suggesting different formation

pathways for nitrate in different seasons. $Na^+$ and $Cl^-$ exhibited dominant unimodal distributions in the range of 2.5–10 μm in autumn, associated with regional transport of sea salt. $Na^+$ and $Cl^-$ size distributions were dominant in the fine mode in winter. OC and EC were generally observed in the fine mode with a comparatively broad size distribution. Our study indicated that vertical meteorological parameters, such as RH and T, and the aqueous and heterogeneous atmospheric chemical reactions altogether led to the aerosol formation and haze episodes in the PRD region. We further simulated the vertical concentration distributions of sulfate, nitrate, and ammonium using WRF-Chem model, and found that the current model could not well reproduce them. Overall, the results of this study can help improve understanding the formation of atmospheric aerosols in polluted sub-tropical environments and can be used to refine global models that simulate the aerosol properties.

**Data availability.** The atmospheric particulate matter data used for analysis are available in the supplementary, and the data are also available upon request from the corresponding author.

**Author contributions.**

SZ and XW designed and led the study. SZ, HZ, JPZ, YC, and JK contributed to aerosols measurement. SZ, JG, and WC carried out the data analysis. LW and WC performed the model simulations. JZ, YFC, YS, PF, and SJ discussed the results and commented on the manuscript. SZ wrote the paper with contributions from all co-authors.

**Competing interests.**

The authors declare that they have no conflict of interest.

**Acknowledgments.**

This work was funded by the National Key Research and Development Program of China (2017YFC0210104, 2016YFC0203305), the National Natural Science Foundation of China (41505106, 41875152, 21577177), the National Science Fund for Distinguished Young Scholars (41425020), the

Fundamental Research Funds for the Central Universities (16lgpy28), and the National Natural Science Foundation as a key project (91644215, 41630422). The authors thank Li Yang, Qi, Zhang, Wenchao Ji, Yuanyuan Jia, Zuozhi Zhong, Hao Liu, Xiaoran, Liu, Xiaodong, Zhang, and Guo Xu for their support in aerosol samples collection and analysis. We also acknowledge the financial support from Guangzhou Environmental Protection Bureau, the State Key Laboratory of Atmospheric Boundary Layer Physics and Atmospheric Chemistry at Institute of Atmospheric Physics.

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

**List of Figure Captions**

**Figure 1.** Map showing the PRD region and Canton Tower sampling sites in Guangdong Province, China. The red contour indicates the spatial distribution of urban areas from global Land Cover data 2015 (www.esa-landcover-cci.org).

**Figure 2.** $PM_{2.5}$ mass and chemical components concentrations at ground level (GND), 118 m and 488 m during the (a) autumn and (b) winter sampling periods. The dates on the x-axis were the sampling days. The stacked bar diagrams for each day represent chemical components at ground level (left), 118 m (middle) and 488 m (right). The green lines represent the daily averaged $PM_{2.5}$ mass concentration. E1 and E2 represent two haze episodes with daily average $PM_{2.5}$ concentrations on the ground site higher than 75 g m$^{-3}$. Error bars represent the standard deviations of the mean.

**Figure 3.** Representative and average vertical profiles of sulfate, ammonium, nitrate, OC, and EC mass concentration at ground level, 118 m and 488 m during (a) autumn; (b) winter. Four layers of $PM_{2.5}$ mass concentrations are shown here with the data measured by Guangzhou EMC. Error bars represent standard deviation of the mean.

**Figure 4.** Relative importance of the main chemical components in (a) fine-mode and (b) coarse-mode particles at three different levels of Canton Tower.

**Figure 5.** Mass concentration size distributions of the main chemical components measured at ground level, 118 m, and 488 m in autumn. The dotted lines represent nonlinear fitting of the measured average size distribution. The error bars represent the sampling and analytical standard errors for each compound.

**Figure 6.** Mass concentration size distributions of the main chemical components measured at ground level, 118 m, and 488 m in winter. The dotted lines represent nonlinear fitting of the measured average size distribution. The error bars represent the sampling and analytical standard errors for each compound.

**Figure 7.** Vertical profiles of relative humidity and temperature modeled by WRF for (a) E1 during autumn and (b) E2 during winter field studies as marked in Fig. 2.

**Figure 8.** Distribution of vertical wind (color scale, red: upward; blue: downward) and cloud fraction (black contour line) simulated by the WRF model during (a) autumn and (b) winter episodes.

**Figure 9.** Schematic graph illustrating a typical haze formation mechanism in the PRD region in autumn and winter.

**Figure 10.** The vertical concentration profiles of sulfate, nitrate, and ammonium in $PM_{2.5}$ during (a) autumn and (b) winter (The red solid lines are the average modeled concentrations and the shaded regions indicate the minimum and maximum values of the simulation; the average measurement data were in black with horizontal error bars).

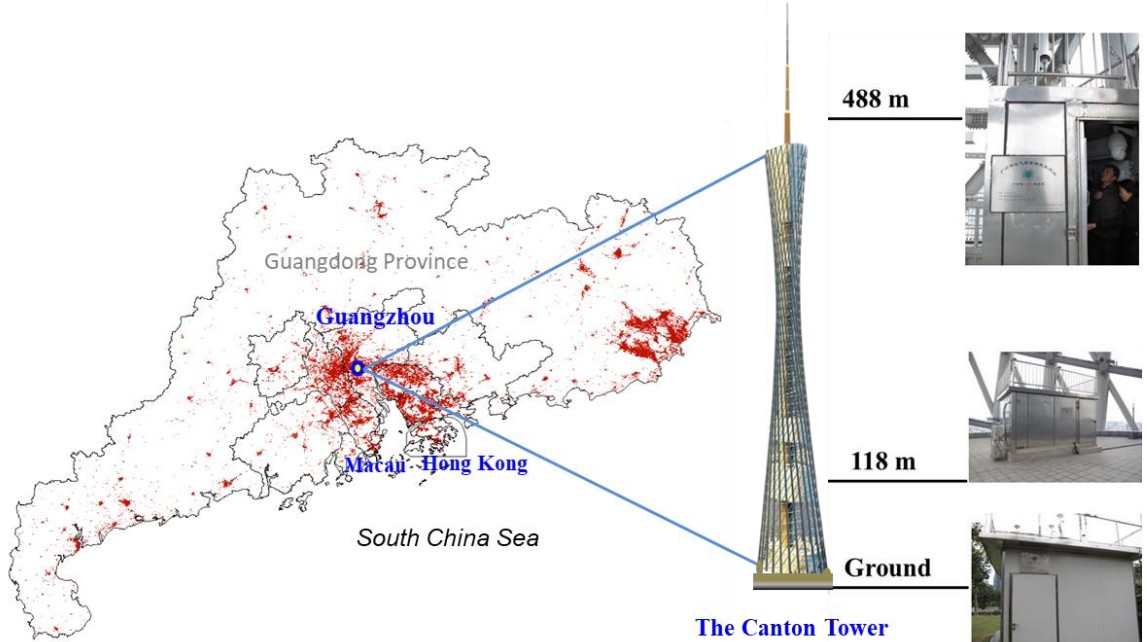

**Figure 1.** Map showing the PRD region and Canton Tower sampling sites in Guangdong Province, China. The red contour indicates the spatial distribution of urban areas from global Land Cover data 2015 (www.esa-landcover-cci.org).

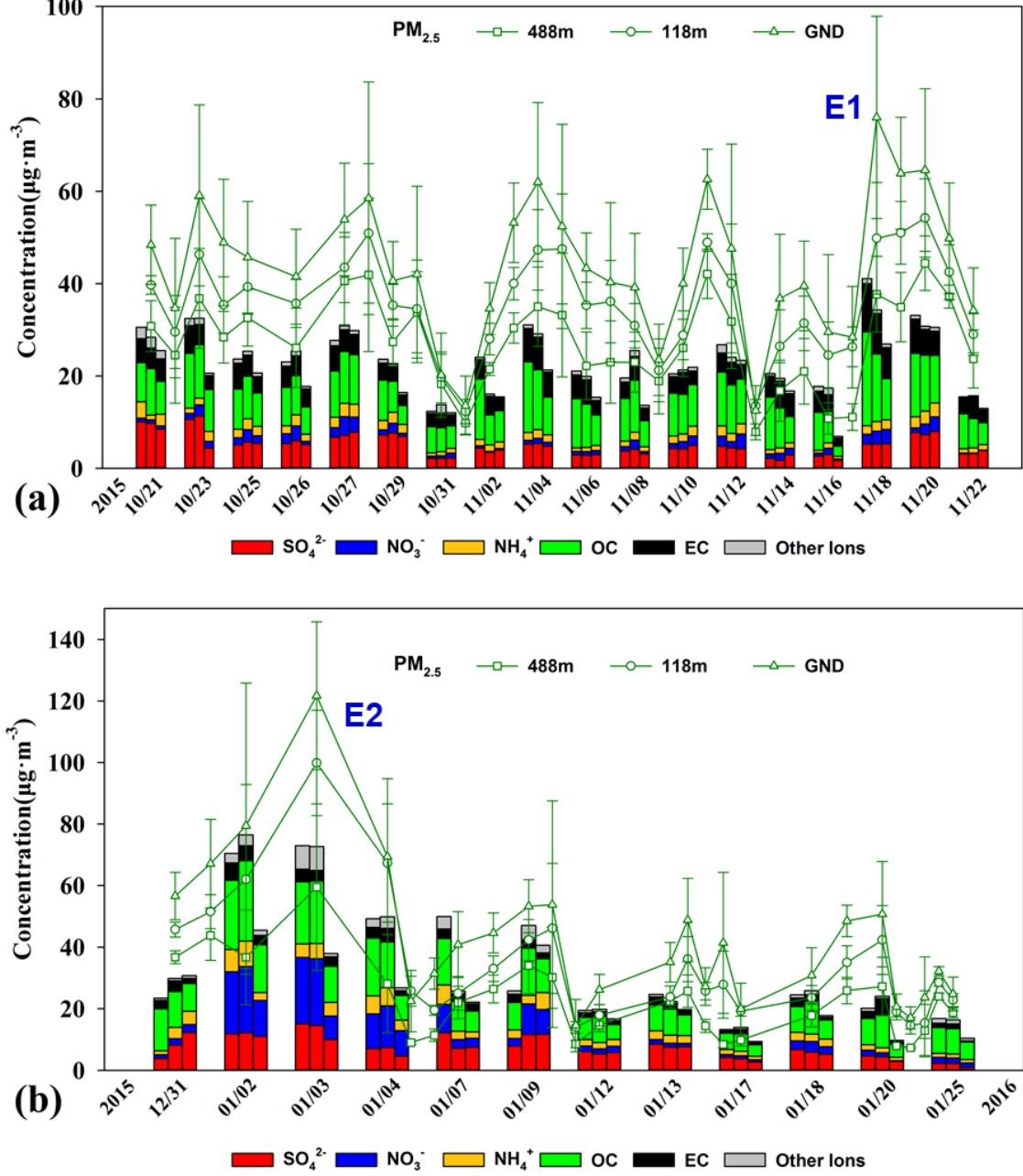

**Figure 2.** PM$_{2.5}$ mass and chemical components concentrations at ground level (GND), 118 m, and 488 m during the (a) autumn and (b) winter sampling periods. The dates on the x-axis are the sampling days. The stacked bar diagrams for each day represent chemical components at ground level (left), 118 m (middle), and 488 m (right). The green lines represent the daily averaged PM$_{2.5}$ mass concentration. E1 and E2 represent two haze episodes with daily average PM$_{2.5}$ concentrations on the ground site higher than 75 μg m$^{-3}$. Error bars represent the standard deviations of the mean.

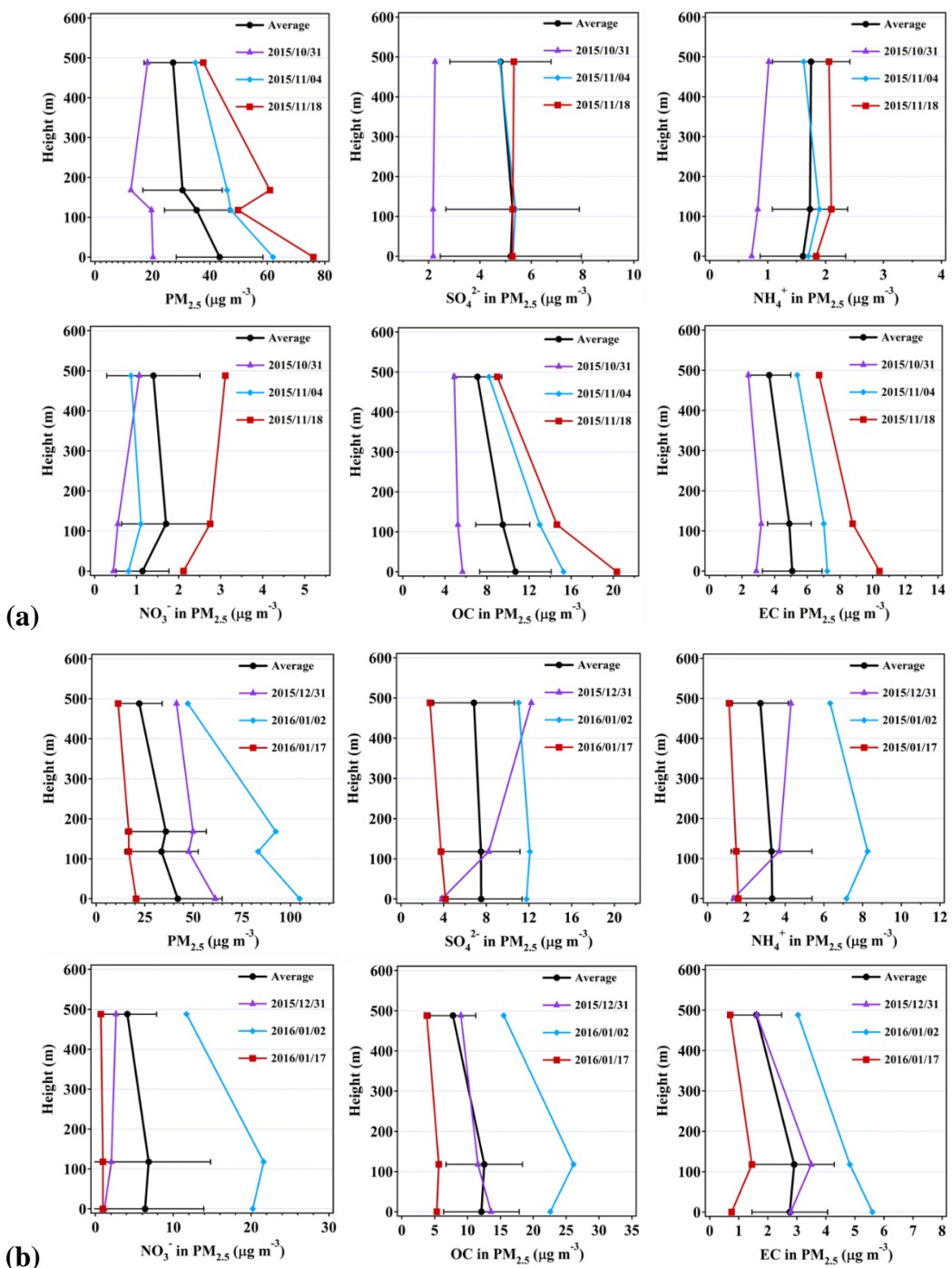

**Figure 3.** Representative and average vertical profiles of sulfate, ammonium, nitrate, OC, and EC mass concentration at ground level, 118 m, and 488 m during (a) autumn; (b) winter. Four layers of $PM_{2.5}$ mass concentrations are shown here with the data measured by the Guangzhou EMC. Error bars represent standard deviation of the mean.

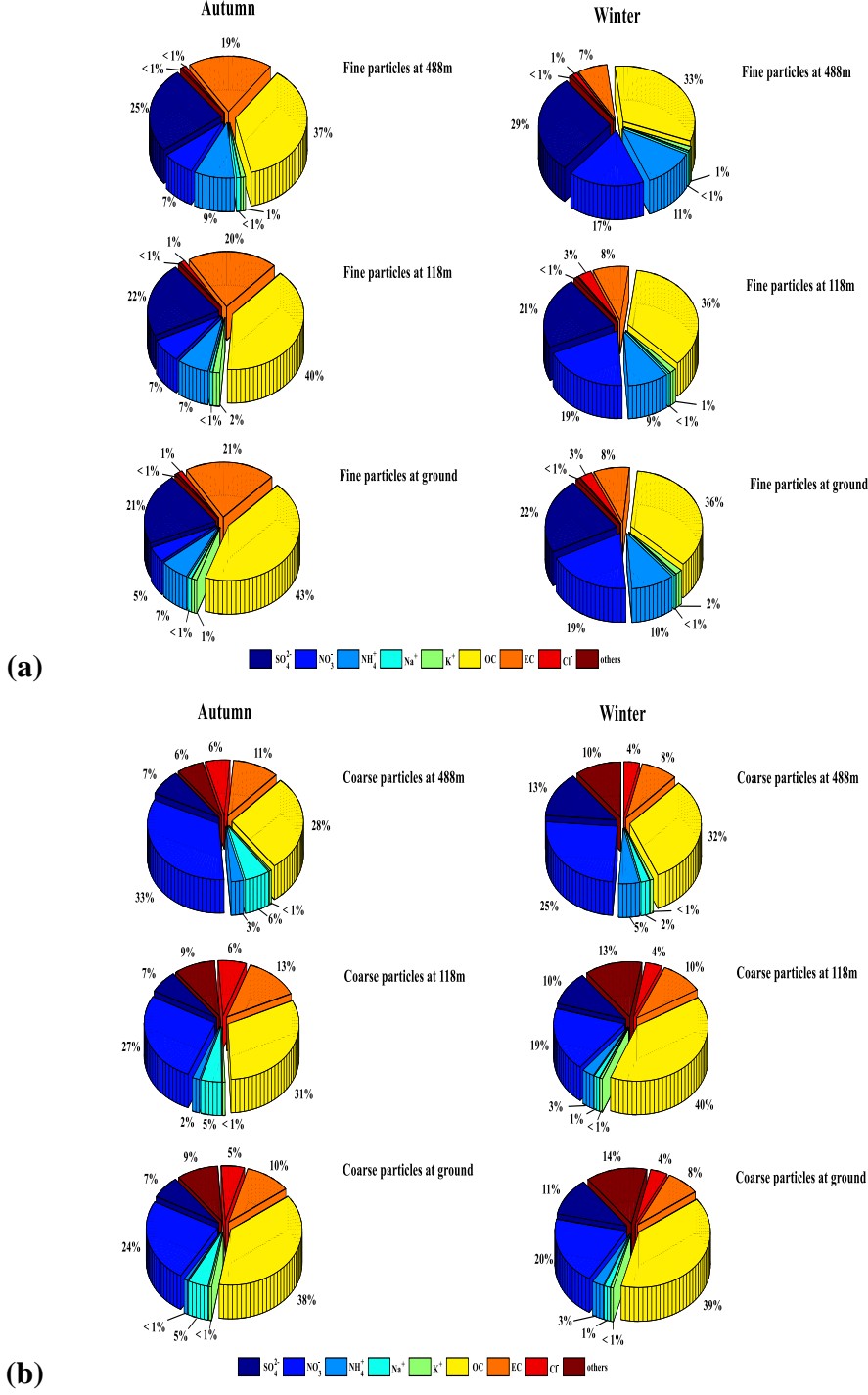

**Figure 4.** Relative importance of the main chemical components in (a) fine-mode and (b) coarse-mode particles at three different levels of Canton Tower.

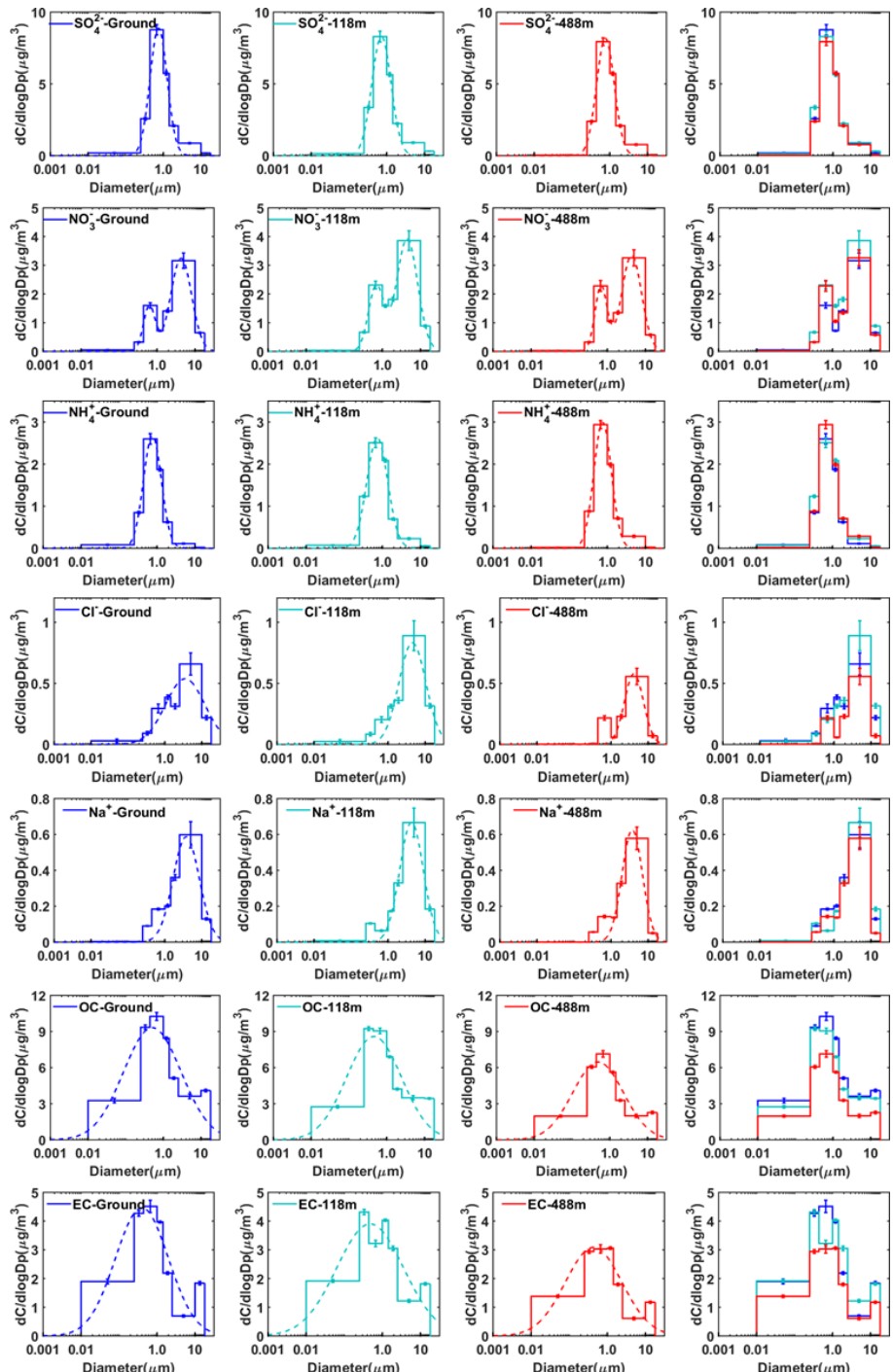

**Figure 5.** Mass concentration size distributions of the main chemical components measured at ground level, 118 m, and 488 m in autumn. The dotted lines represent nonlinear fitting of the measured average size distribution. The error bars represent the sampling and analytical standard errors for each compound.

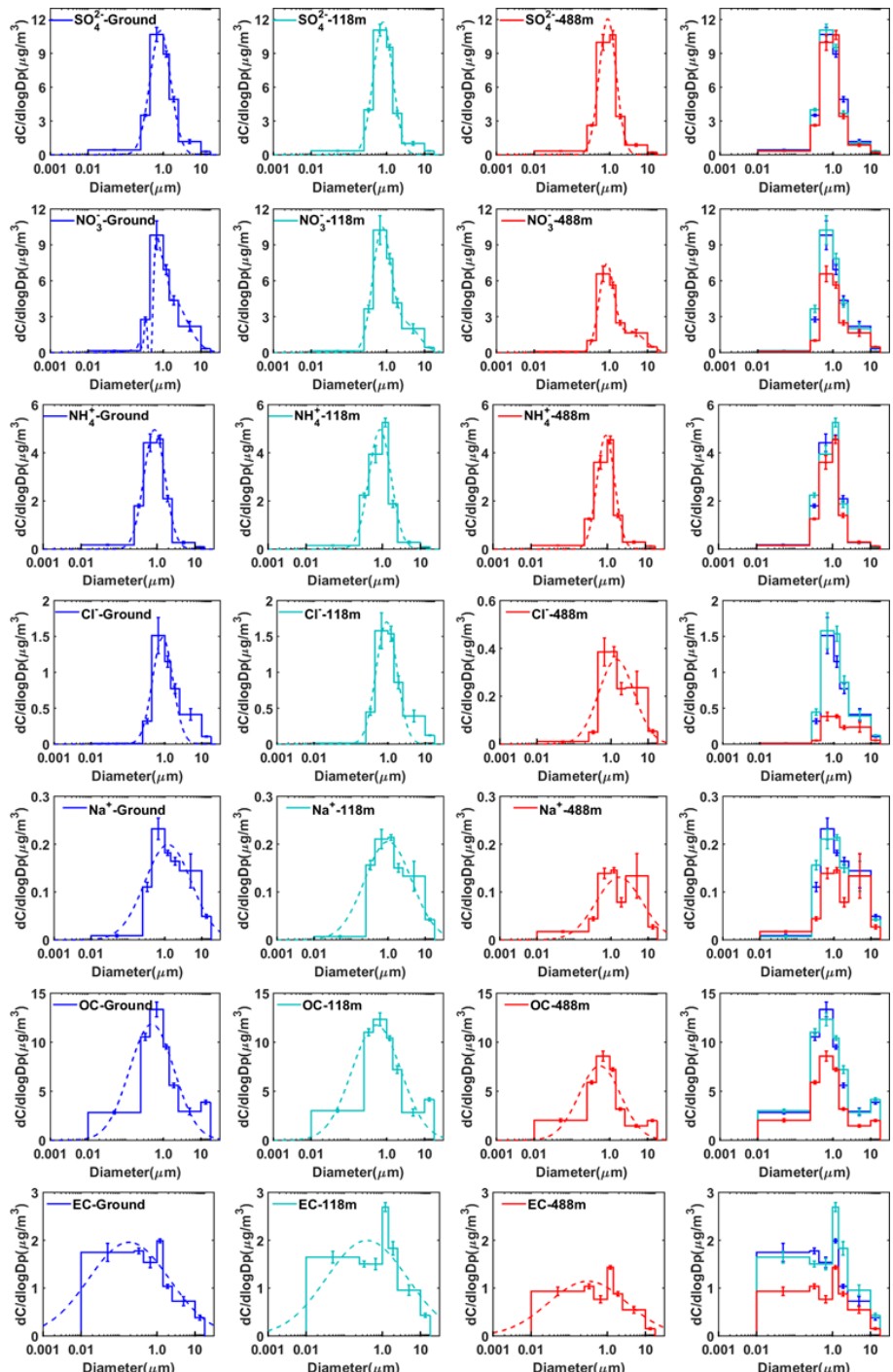

**Figure 6.** Mass concentration size distributions of the main chemical components measured at ground level, 118 m and 488 m in winter. The dotted lines represent nonlinear fitting of the measured average size distribution. The error bars represent the sampling and analytical standard errors for each compound.

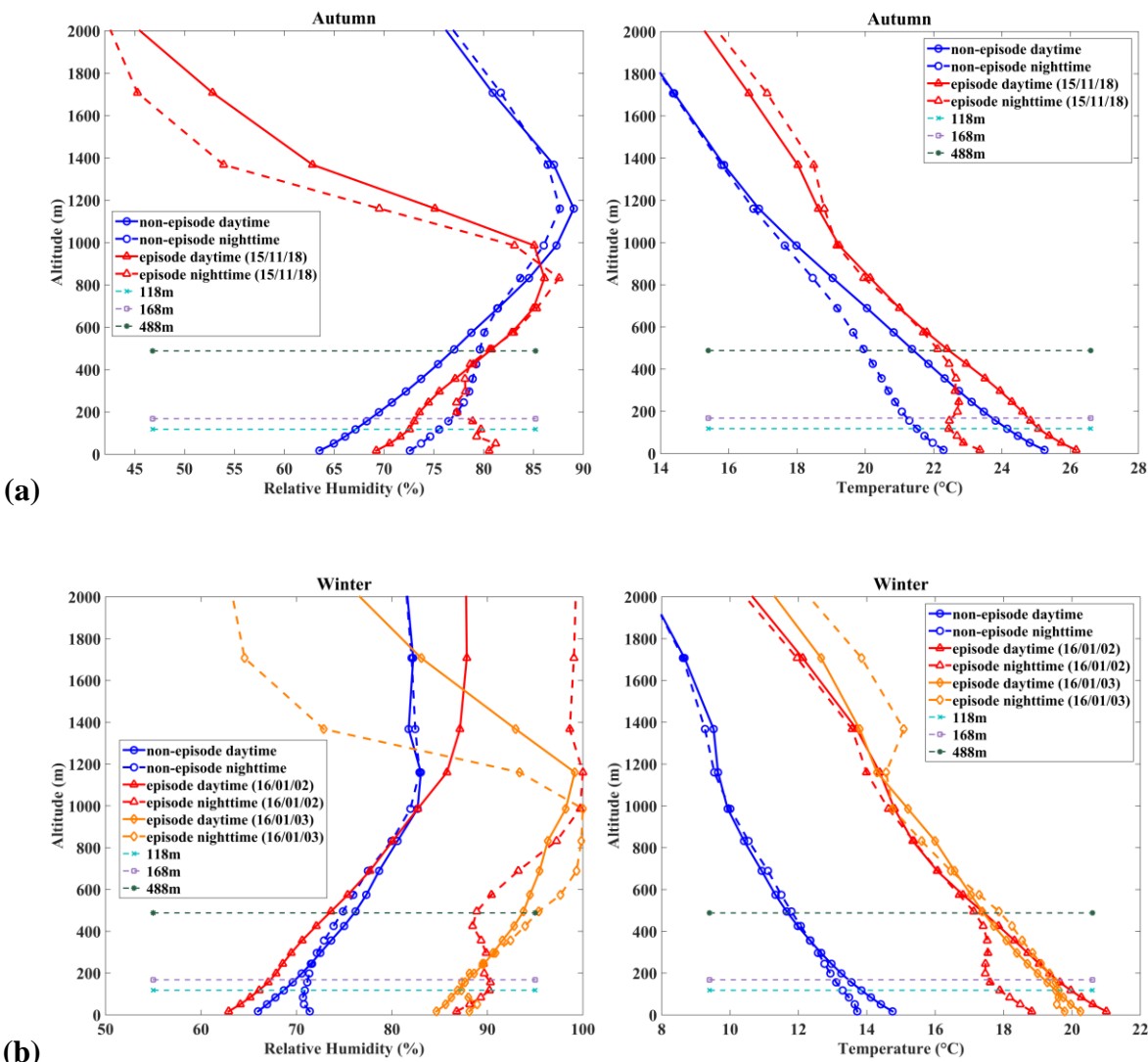

5 **Figure 7.** Vertical profiles of relative humidity and temperature modeled by WRF for (a) E1 during autumn and (b) E2 during winter as marked in Fig. 2.

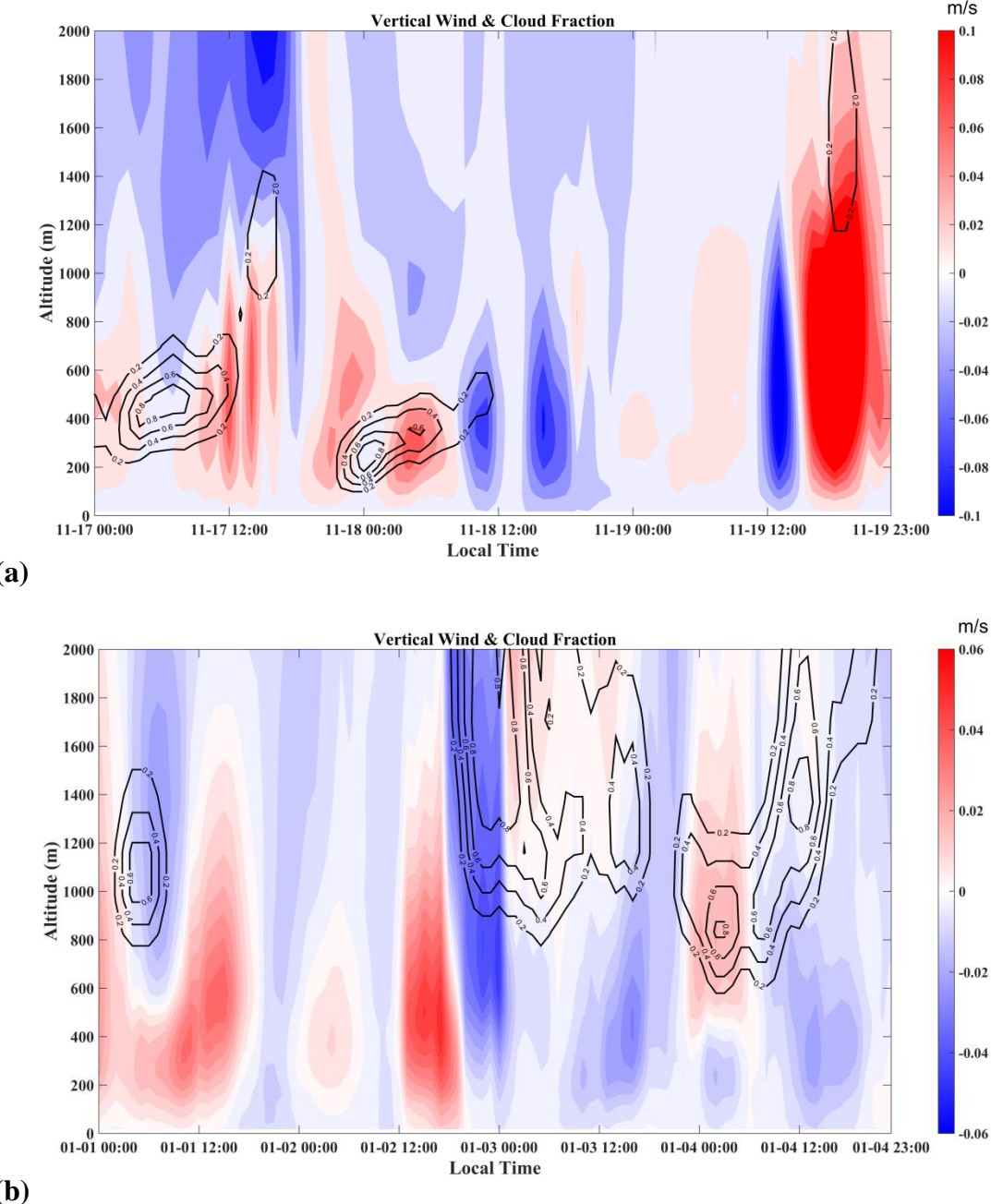

**Figure 8.** Distribution of vertical wind (color scale, red: upward; blue: downward) and cloud fraction (black contour line) simulated by the WRF model during (a) autumn and (b) winter episodes.

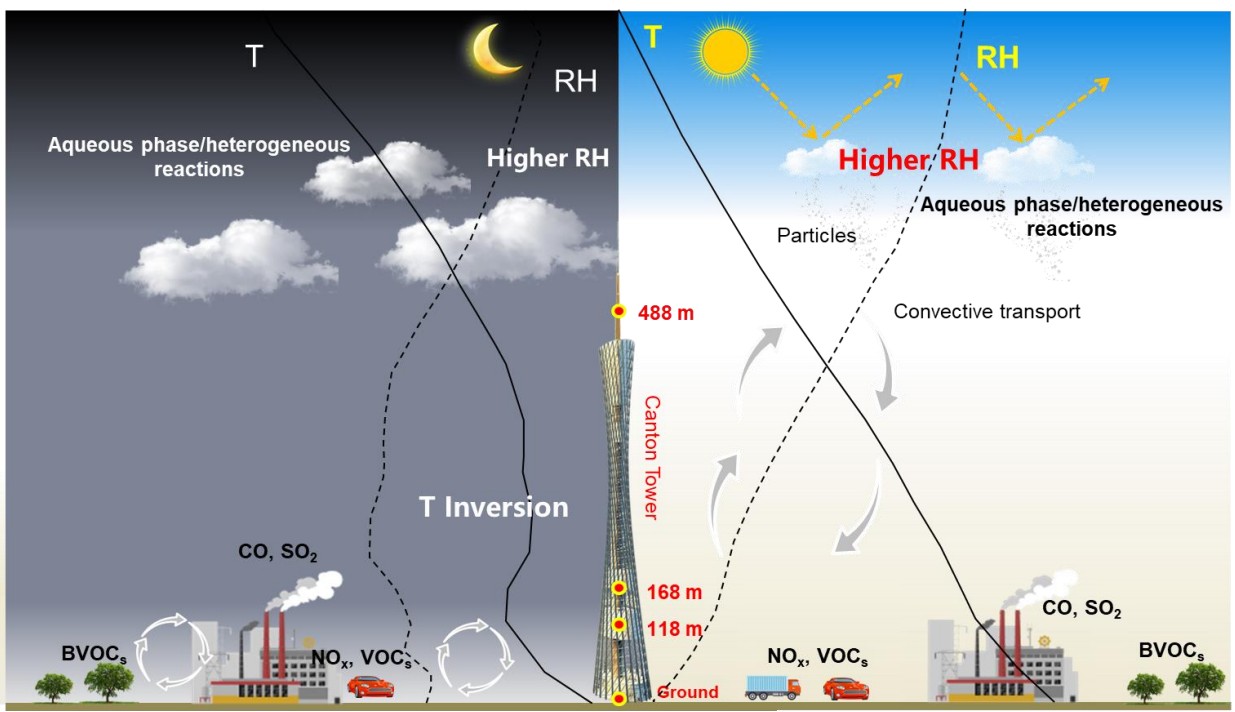

**Figure 9.** Schematic graph illustrating typical haze formation mechanisms in the PRD region in autumn and winter.

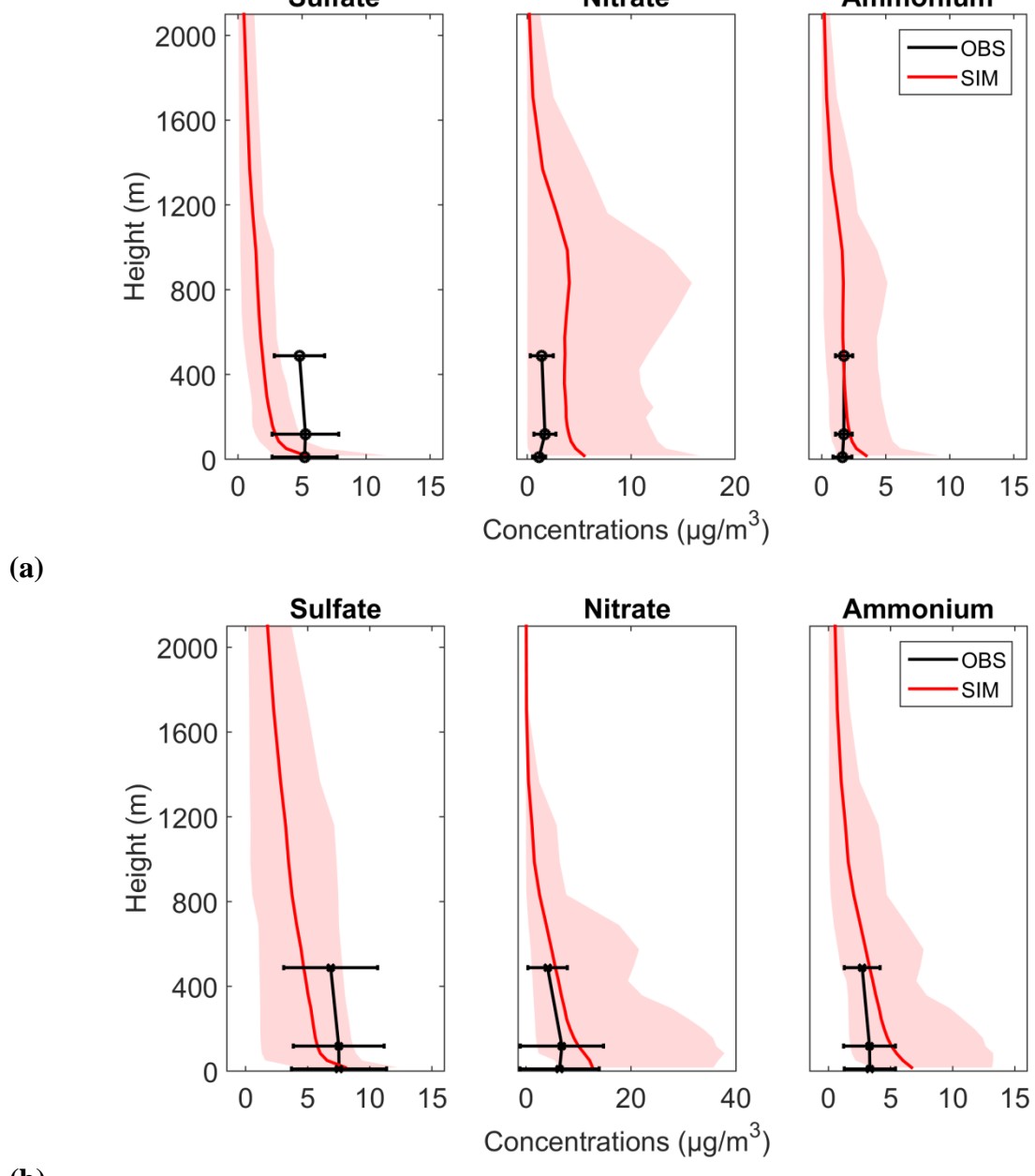

**(a)**

**(b)**

5   **Figure 10.** The vertical concentration profiles of sulfate, nitrate, and ammonium in PM$_{2.5}$ during (a) autumn and (b) winter

(The red solid lines are the average modeled concentrations and the shaded regions indicate the minimum and maximum

values of the simulation; the average measurement data were in black with horizontal error bars).