# Peer review of "Vertical distribution of atmospheric particulate matter within the urban boundary layer in southern China: Size-segregated chemical composition and secondary formation through cloud processing and heterogeneous reactions"

_Atmospheric Chemistry and Physics, 2019_

## Referee Comment (RC1) · Anonymous Referee #1 · 7 May 2019

The study by Zhou et al. is an interesting one considering the fact that very high towers are not readily accessible to researchers. Having said that, similar studies have been performed around the world and despite claiming the uniqueness of the study it is not entirely unique, except for utilising perhaps the highest tower. However, it is not the height of the tower which makes any study unique, but instead scientific insights about

the processes drawn from it. The study is not without significant drawbacks and needs significant improvement to warrant publication in the respected journal of Atmospheric Chemistry and Physics. Last but not least, English of the manuscript needs significant improvement as many sentences are unclear or dubious.

Major comments

The introduction needs significant improvement as overall interpretation of PM sources and processes is rather outdated, or straightforward or not consistent with the most recent fundamental papers. Very often the authors choose to reference either old papers missing out on recent ones, or choose to reference very recent, neglecting pioneering earlier papers. It is unclear what exactly were the goal and aims of the study other than utilising a very high tower. Those goals should normally arise from the earlier papers by identifying scientific knowledge gaps and which the authors choose to advance upon.

The paper currently stands more like a report rather than a scientific paper. It presents data, but lacks coherent view. More often than not the authors seek consistency with other studies or providing references which support or fit their data. Taken altogether, the paper is currently a collection of interpretations which are not always consistent with each other and most importantly lacking conclusive findings which would advance rather than confirm already known processes or phenomena.

The study is lacking an overview of all the profiles, splitting into certain categories and introducing the scope and variability of the data set first. There is a complete lack of uncertainty and error analysis. Analytical and sampling uncertainty would propagate into vertical profile uncertainties which would then make profiles or concentrations at different heights significantly different or not. The authors choose to select specific episodes or profiles which are most obvious or interpretable and neglecting which are not. Selected profiles are certainly insightful, but only when put into overall context.

Considering the challenges in organising such a study it is pity that high frequency micrometeorological measurements (including an important vertical wind speed) were not undertaken making it impossible to derive fluxes (refer to papers by (Valiulis, Ceburnis et al. 2002, Ceburnis, O'Dowd et al. 2008).

Minor comments

Page 2, line 27. Outdated literature overview of the processes involved and oversimplifying the system.

Page 3, line 5. Old literature support. There is plenty of hard evidence that the first sub-mode can also be formed by cloud processing, e.g. (Ovadnevaite, Zuend et al. 2017)

Line 6. Unclear sentence - what was exactly demonstrated worldwide?

Line 9. Sea salt can also be submicron down to 20nm, e.g. (Ovadnevaite, Manders et al. 2014, Cravigan, Ristovski et al. 2015)

Line 19. One of the earliest papers published by (Valiulis, Ceburnis et al. 2002) which also estimated emissions from the observed gradients.

Page 4, line 13. Introduction should only present goals and objectives of the study and not the description of measurements performed unless nobody measured them before which is not the case here.

Line 26. delete "the other three levels".

Page 5, line 5. Why 168m level is missing? 168m is often missing in the results section and is not clearly explained why.

Page 6, line 17. Why the study is focused only in several pollution episodes when an overview of gradient should be presented first including error bars and uncertainties. Selected gradients discussed later become suspicious whether they are representative or just being random.

[Figure]

Line 23. It is not the consistency with other studies that makes the measurements reliable trusted. Instead, decreasing concentration with height points at the ground/surface sources as opposed to increasing concentration with height pointing at sources aloft (Ceburnis, O'Dowd et al. 2008)

Page 7, line 1. This needs to be investigated if not occurring due to temperature inversions impeding mixing. That can be especially true in winter, but temperature inversions readily forming under clear sky condition due to radiative cooling.

Line 6. "Concentration gap" is unclear and unsuitable term.

Line 10. Composition cannot be vertically distributed - chemical components are distributed instead.

Line 16. Repetition.

Line 21. It does not need to be associated to EC as many other species are emitted by sources at the ground.

Line 24. Please spell acronyms used for the first time.

Line 27. No established vertical gradients...

Page 8, line 12. Secondary WSOC formation is the scientific fact - why is it missing from interpretation?

Line 21. If distributions were averaged they must be presented with errors bars or ranges.

Line 27. Sulfate having similar formation mechanism to what?

Page 9, line 4. Coagulation is negligible at typical ambient number concentrations (refer to e.g. W.C. Hinds Aerosol Technology Textbook). In-cloud coalescence of droplets is more likely or multiple cloud cycles could explain production of several modes.

Line 20. Formation of nitrate is not exclusive to sea salt or dust particles, any surface

would promote heterogeneous reactions.

Page 11, line 5. Chloride particles do not exist and chloride cannot be considered separately from sodium or other balancing ion like ammonium.

Line 13. Incorrect suggestion. That finding is only demonstrating aged aerosol which undergone significant chemical processing during advection to the sampling location.

Line 22. Unclear sentence. Similar origin of OC at three heights? Its unlikely as similar concentrations can be produced by proportional contribution of ground sources versus in-cloud processing.

Line 27. That is not a possibility, but rather the only plausible explanation. However, the fact that nearby chimneys may have affected the profiles diminishes the value of this study making the interpretation of profiles very speculative and simply fitting the observations.

Page 12, line 21. Observed, not found.

Line 28. Why the authors suggest what was already pointed out as contribution from nearby chimneys and stacks?

Page 13, line 10. ...temperature inversion extending from 118 to 488 meters...

Line 16-23. Schematics is haphazardly constructed and needs much better discussion and reasoning based on observations. 8 lines are absolutely insufficient. This schematics should be significantly improved or removed altogether. Figure 9. Why an upward convective transport missing? Schematics is lacking sampling heights to validate the processes.

Line 26. "Utilizing the 610m Canton Tower in Guangzhou" has to be moved to the sentence end.

Line 28. Small or smaller? Shallower is perhaps the better word.

Page 14, line 6. OC missing

Line 15. In order for the results of the study helping understanding formation mechanisms, the data interpretation needs considerable improvement.

Ceburnis, D., C. D. O'Dowd, G. S. Jennings, M. C. Facchini, L. Emblico, S. Decesari, S. Fuzzi and J. Sakalys (2008). "Marine aerosol chemistry gradients: Elucidating primary and secondary processes and fluxes." Geophysical Research Letters 35(7): L07804.

Cravigan, L. T., Z. Ristovski, R. L. Modini, M. D. Keywood and J. L. Gras (2015). "Observation of sea-salt fraction in sub-100 nm diameter particles at Cape Grim." Journal of Geophysical Research: Atmospheres 120(5): 1848-1864.

Ovadnevaite, J., A. Manders, G. de Leeuw, D. Ceburnis, C. Monahan, A. I. Partanen, H. Korhonen and C. D. O'Dowd (2014). "A sea spray aerosol flux parameterization encapsulating wave state." Atmospheric Chemistry and Physics 14(4): 1837-1852.

Ovadnevaite, J., A. Zuend, A. Laaksonen, K. J. Sanchez, G. Roberts, D. Ceburnis, S. Decesari, M. Rinaldi, N. Hodas, M. C. Facchini, J. H. Seinfeld and C. O. Dowd (2017). "Surface tension prevails over solute effect in organic-influenced cloud droplet activation." Nature 546(7660): 637-641.

Valiulis, D., D. Ceburnis, J. Sakalys and K. Kvietkus (2002). "Estimation of atmospheric trace metal emissions in Vilnius City, Lithuania, using vertical concentration gradient and road tunnel measurement data." Atmospheric Environment 36(39-40): 6001-6014.

---

## Referee Comment (RC2) · Anonymous Referee #2 · 21 May 2019

This manuscript presents measurements of the vertical distribution of aerosol composition in Guangzhou, China and, based on those data, provides insight into the sources and formation mechanisms of the different chemical components. The manuscript is certainly understandable, but would require editing prior to publication. As is noted in the paper, datasets such as these can be very useful and yet there are very few available. Because of that utility I feel that this dataset should be published. However, I feel that the use of the data to infer aerosol sources is weak and can probably not be refined to the level that would make ACP the appropriate journal for publication.

Simultaneous and semi-continuous measurements such as those described here are challenging to make and are more amenable to collection and off-line analysis techniques such as those used here. Nevertheless, interpretation of the data is constrained by the resulting low time (24 h) and size (7 bins) resolution. For example, some of the central conclusions of the manuscript are based on the relative concentrations of species in the droplet mode, but that mode is contained in a very broad $0.44 - 1.0$ micron bin. The time resolution also complicates the interpretation, especially when attempting to connect the measurements with meteorology in the case studies.

I have questions/concerns about the impactors, in part because I have never used these and am unfamiliar with aspects of their operation. i) How were they calibrated? There is no information provided and the calibrated (?) cut sizes just happen to be exactly the same as those reported by the manufacturer. ii) How is the flow rate controlled? I ask because any pressure and/or temperature dependence would influence the recorded concentration height dependence. And connected to that, are the reported concentrations at local temperature and pressure or are they adjusted to standard (or other) conditions? iii) Is the air dried upstream of the impactors? For such a large flow rate I suspect the answer is no. And if not, this could have important impacts on the relative size distributions of the different chemical species and of the same species at different heights. The size distributions of hygroscopic species such as sulfate and nitrate would be shifted, while those of less- or non-hygroscopic species such as OC and EC would not. I appreciate that such shifts are not solely responsible for the differing size distributions, but they could be a contributor. The size distribution of those hygroscopic species would also vary with height due to variation in RH. This might partly explain the observation that the sulfate size distribution was shifted more into the droplet mode at 488 m, which was argued to be evidence of cloud processing in the manuscript (page 9, line 10). The average RH of between 78% and 80% suggests the bias could be significant.

The authors interpret the relatively flat vertical profile of sulfate compared with EC and OC as an indication of the importance of cloud processing. But there is no explanation provided about how that contrast would differ for local emissions of EC and OC and simply regional production of sulfate (gas or aqueous phase). I'm not so sure the difference would be easily discerned.

I agree that using meteorology to interpret the aerosol data and to constrain the origin and formation of the different species is logical. But I don't agree that almost exclusively relying on WRF model output is reasonable. Why not at least validate those elements of the model predictions for which surface and/or satellite observations are available. Cloud cover and cloud base height are two that come to mind.

For much of the discussion I believe it would be better to describe variations in absolute rather than relative concentrations. I recognize that for some explanations it is appropriate to describe differences in percent contribution of one or more species to the total concentration. But for other descriptions varying concentrations of other species unnecessarily complicates the results. One example is the conclusion on page 8, line 12 of favorable formation of the inorganic ion species based on relative changes in composition with height. The percentages would of course change in the same way if the concentrations of those species increased or those of other species decreased.

Minor issues in the order in which they appear in the manuscript:

Page 3, Line 8: What are irregular sizes? And dust is usually used instead of sand.

Page 7, Line 24: SNA spelled out only later.

Figure 2: Some explanation should be provided for the mismatch between the real-time and offline estimates of PM2.5.

Figure 5: Some explanation should be provided for why the concentration of NH4 is

highest at 488 m while that of SO4 and NO3 are not.

Figures 5 and 6: I believe the 0.1 on all of the x-axes is supposed to be 0.01 and 0.1 is for some reason not shown.

---

## Author Comment (AC1) · 2 Aug 2019

The comment was uploaded in the form of a supplement:
https://www.atmos-chem-phys-discuss.net/acp-2019-155/acp-2019-155-AC1-supplement.pdf

---

## Author Comment (AC2) · 2 Aug 2019

Response to the referee comments

**Response to Anonymous Referee #2**

This manuscript presents measurements of the vertical distribution of aerosol composition in Guangzhou, China and, based on those data, provides insight into the sources and formation mechanisms of the different chemical components. The manuscript is certainly understandable, but would require editing prior to publication. As is noted in the paper, datasets such as these can be very useful and yet there are very few avail-able. Because of that utility I feel that this dataset should be published.

**[A]**: We would like to express our sincere appreciation for the reviewer's careful reading and invaluable comments to improve the paper. We have revised the manuscript accordingly. Please kindly find our itemized responses to the specific comments below. The reviewer's comments are in black and the authors' responses are in blue. Any changes made in the revision are highlighted in red.

**[1]** However, I feel that the use of the data to infer aerosol sources is weak and can probably not be refined to the level that would make ACP the appropriate journal for publication.

**[A]:** We thank the reviewer for valuable suggestions. In this study, we are not aiming at quantitively source apportionment but more to gain insight on the possible processes and mechanisms (e.g., secondary formation through cloud processing and heterogeneous reactions) that could contribute to the particulate matter formation with detailed mass size-resolved chemical components of particulate matters at different vertical levels. To clarify this, we have thoroughly revised the introduction.

Page 4, lines 14-28 and page 5, lines 1-3: "Severe aerosol pollutions frequently occur in China, as exemplified by three cities groups in the Jing-Jin-Ji (Beijing, Tianjin, and Hebei province), the Yangtze River Delta, and the Pearl River Delta regions. State–of–the–art air quality models still often fail to simulate the observed high $PM_{2.5}$ concentrations even after including aerosol-radiation-meteorology feedback, indicating that key atmospheric chemical processes, such as heterogeneous and

multiphase reactions, are lacking in models for secondary aerosol formation (Zheng et al., 2015; Cheng et al., 2016). To improve the understanding of haze formation, models will require updated kinetic and mechanistic data of multiphase chemistry and quantification of the aerosol formation through heterogeneous reactions under real atmospheric conditions (Zheng et al., 2015; An et al., 2019). Additionally, more consistent evidences of aerosol formation through heterogeneous reactions are needed from field measurements, laboratory experiments and model simulations. Field studies showed that extremely high $PM_{2.5}$ concentrations usually occurred under high relative humidity conditions (Sun et al., 2014; Wang et al., 2014). Heterogeneous aqueous phase reactions in the cloud liquid water and in aerosol water can promote secondary aerosol formation (Seinfeld et al., 2006; Ervens, 2015; McNeill, 2015; Cheng et al., 2016). It is hence critical to investigate the aerosol sources and formation mechanisms by measuring size-resolved PM components vertically using a tall tower, where they can be strongly influenced by the dynamic variations of atmospheric boundary layer and cloud processing."

In addition, we have clarified the objectives of this study in the revised manuscript.

Page 5, lines 13-17: "The objectives of this study are to (1) analyze the vertical mass size distribution of the PM chemical components and the factors that affect their vertical variations; and (2) investigate the roles of in-cloud processes and heterogeneous aqueous reactions in secondary aerosol formation and the implication for haze pollution in subtropical urban areas."

[2] Simultaneous and semi-continuous measurements such as those described here are challenging to make and are more amenable to collection and off-line analysis techniques such as those used here. Nevertheless, interpretation of the data is constrained by the resulting low time (24 h) and size (7 bins) resolution. For example, some of the central conclusions of the manuscript are based on the relative concentrations of species in the droplet mode, but that mode is contained in a very broad 0.44 – 1.0 micron bin. The time resolution also complicates the interpretation,

especially when attempting to connect the measurements with meteorology in the case studies.

[A]: We thank the reviewer for valuable comments and suggestions. We agree that our measurements were low in time and size resolution due to the limited instruments and sampling site during the study. It is really difficult to conduct measurements in the vertical direction on Canton tower, and we've done our best so far. However, although size and time resolutions were low, the results from our study can still provide useful aerosol mass size distribution which is complimentary to the number size distribution usually measured in other field studies. Our results can provide a general characteristic of air pollutions in the PRD region, and useful information on the aerosol sources and transformations by the direct aerosol mass size measurements. We will plan to involve high time and size resolution measurements in the future study.

[3] I have questions/concerns about the impactors, in part because I have never used these and am unfamiliar with aspects of their operation. i) How were they calibrated? There is no information provided and the calibrated (?) cut sizes just happen to be exactly the same as those reported by the manufacturer. ii) How is the flow rate controlled? I ask because any pressure and/or temperature dependence would influence the recorded concentration height dependence. And connected to that, are the reported concentrations at local temperature and pressure or are they adjusted to standard (or other) conditions? iii) Is the air dried upstream of the impactors? For such a large flow rate I suspect the answer is no. And if not, this could have important impacts on the relative size distributions of the different chemical species and of the same species at different heights. The size distributions of hygroscopic species such as sulfate and nitrate would be shifted, while those of less- or non-hygroscopic species such as OC and EC would not. I appreciate that such shifts are not solely responsible for the differing size distributions, but they could be a contributor. The size distribution of those hygroscopic species would also vary with height due to variation in RH. This might partly explain the observation that the sulfate size distribution was shifted more into the droplet mode at 488 m, which was argued to be evidence of

cloud processing in the manuscript (page 9, line 10). The average RH of between 78% and 80% suggests the bias could be significant.

[A]: We thank the reviewer for insightful comments. We give an overview of the impactor and then address the reviewer's comments point-by-point as shown below. The High-Flow Impactor (Model 131) is a commercial aerosol sampler manufactured by MSP corp. in USA. Its operation principle is inertial impaction using multiple-nozzle stages in series. At each stage, particle-laden air jets impinge upon an impaction plate. Particles larger than the cut-size of that stage cross the flow streamlines and are collected on the impaction plate below the nozzles. Particles smaller than the cut-size can follow the flow streamlines and proceed on to the next stage where the nozzles are smaller, the air velocity through the nozzles is higher and the cut-size is smaller. This continues on through the cascade impactor until the smallest particles which are not able to impact on the last impaction plate are collected by a final filter.

The Model 131 High-Flow Impactor consists of an inlet (which is also a large-particle pre-separator), six impaction stages and a filter holder-base. Stages 1 through 6 and the filter holder-base support the removable 75-mm impaction plates for the pre-separator inlet and stages 1 through 6 respectively. Figure I shows the Model 131 High-Flow Impactor we used in the samples collection.

[Figure]

Figure I. Model 131 High-Flow Impactor

**i)** How were they calibrated? There is no information provided and the calibrated (?) cut sizes just happen to be exactly the same as those reported by the manufacturer.

➢ We have added more information on the flow rate calibration in the revised manuscript.

Page 6, lines 5-13: "Three impactors (or samplers) were calibrated using mass flow meter (TSI, model 4040) in the laboratory before they were used during the study. The flow rates of the impactors were measured at the beginning of the sampling. At the end of the sampling period, the flow rates were recorded again. If the flow rate of each impactor at the beginning and end of the sampling period differed by more than 10%, the sample was marked as suspect and the data was discarded. The average flow rates at the beginning and end of the sampling time was used as the sampling flow rate. In addition, a magnehelic pressure gauge was used to monitor the inlet flow rate through the impactor. The pressure drop was also recorded at the beginning and end of sampling."

➢ These cut-sizes are based on the application of current impactor theoretical predictions (Rader and Marple, 1985). Figure II shows the calibration efficiency curves of the five standard impaction stages by the manufacture. These curves have been fitted with cumulative lognormal distributions to determine the calibration cut-size and geometric standard deviation for each stage. Based on the principle, the cut sizes should be the same as those reported by the manufacturer.

[Figure]

Figure II. High-Flow impactor efficiency curves provided by the manufacture

**Reference:**

Rader, D. J. and Marple, V. A., "Effect of Ultra-Stokesian Drag and Particle Interception on Impaction Characteristics" Aerosol Science and Technology, 4: 141-156, 1985.

**ii)** How is the flow rate controlled? I ask because any pressure and/or temperature dependence would influence the recorded concentration height dependence. And connected to that, are the reported concentrations at local temperature and pressure or are they adjusted to standard (or other) conditions?

➢ The magnehelic pressure gauge can be used to monitor the inlet flow rate through the impactor (each impactor has its own special pressure drop at 100 L/min). The low pressure side of the gauge is connected to the pressure tap on the last impactor stage body. The exhaust port of the impactor is connected to the suction side of a suitable vacuum pump. **A flow control** valve is applied to adjust the impactor inlet flow rate to 100 L/min.

We actually did not adjust to standard conditions given the vertical height is less than 500 m and the impacts are small. To prove this, we calculated the impacts of pressure and temperature on the flow rate, and found that these impacts were less than 5%. Blow are our simple calculations based on the measurements of relevant parameters on the Canton tower on Oct. 23, 2015:

The daily average temperatures were 28.0 $^{\circ}$C and 24.1 $^{\circ}$C at the ground level and 488 m, respectively. And the daily average atmospheric pressures were 101.15 kPa and 95.72 kPa at these two levels. The flow rate at the ground level is 100 L/m$^3$. We calculated the flow rate when the temperature was 24.1 $^{\circ}$C and the pressure was 95.72 kPa, i.e. at 488 m, assuming a flow rate of 100 L/m$^3$ at the ground level (temperature = 301.15 K and pressure = 101.15 kPa).

Assume the ambient air is an ideal gas. At the ground level, $P_1$ = 101.15 kPa, $V_1$= 100 L/m$^3$, $T_1$= 273.15+28=301.15 K. At 488 m, $P_2$ = 95.72 kPa, $V_2$= ?, $T_2$= 273.15+24.1=297.25 K. $R$ is the ideal gas constant. $n$ is the moles of air.

We obtain: $P_1 V_1 = nRT_1$ $\qquad$ (1)

$\qquad\qquad P_2 V_2 = nRT_2$ $\qquad$ (2)

(1)/(2) we get:

$$V_2 = \frac{P_1 V_1 T_2}{P_2 T_1} = \frac{101.15 \times 100 \times 297.25}{95.72 \times 301.15} = 104.3 L/m^3$$

We conclude that the impacts of pressure and temperature on the flow rate are less than 5%. **Therefore, we thought that no adjustment of the flow rate was needed.**

**iii)** Is the air dried upstream of the impactors? For such a large flow rate I suspect the answer is no. And if not, this could have important impacts on the relative size distributions of the different chemical species and of the same species at different heights……

➢ We thank the reviewer for the valuable comments. The air was not dried upstream of the impactor in our measurement. We agree that the shifts of the particles sizes would happen due to the increase of relative humidity. However, we think this influence is unlikely to change our conclusion on the droplet mode. Meng and Seinfeld (1994) have proved that water accretion alone cannot account for the growth of droplet-mode particles from the condensation mode. They therefore proposed that activation of condensation mode particles to form fogs or clouds followed by aqueous-phase sulfate formation (also for nitrate and ammonium) and fog evaporation is shown to be a plausible mechanism for formation of the urban and regional aerosol droplet mode. Their findings support our results that in-cloud processing is likely an important source for droplet mode aerosols.

To clarify, we add these discussions into the manuscript as a caveat.

Page 11, lines 9-17: "Relative humidity would influence the relative size distributions of the different chemical species. The air was not dried upstream of the impactor in our measurement. However, we think this influence is unlikely to change our conclusion on the droplet mode. Meng and Seinfeld (1994) have proved that water accretion alone cannot account for the growth of droplet-mode particles from the condensation mode. They proposed that activation of condensation mode particles to form fogs or clouds followed by aqueous-phase sulfate formation (also for nitrate and

ammonium) and fog evaporation are shown to be a plausible mechanism for formation of the urban and regional aerosol droplet mode. Their findings support our results that in-cloud processing is likely an important source for droplet mode aerosols."

Reference:
Meng, Z. and Seinfeld, J.H. On the source of the submicrometer droplet mode of urban and regional aerosols. Aerosol Science and Technology, 20(3): 253-265.

**[4]** The authors interpret the relatively flat vertical profile of sulfate compared with EC and OC as an indication of the importance of cloud processing. But there is no explanation provided about how that contrast would differ for local emissions of EC and OC and simply regional production of sulfate (gas or aqueous phase). I'm not so sure the difference would be easily discerned.

[A]: We have modified the section "3.2.1 Vertical distribution of the major chemical components." (page 8, lines 19-28 and page 9).

We performed the 72-h back-trajectory analysis in Figure S3 in the supplementary. We found that the air masses mainly came from either local or from the South China Sea. From the previous study, we know that the PRD region is one of the air pollution hot spot (Figure III). Therefore, local emissions may contribute significantly to the air quality of the PRD region during the sampling periods.

[Figure]

(a)

[Figure]

**(b)**

Figure S3. Cluster analysis of the airflow in 200 m and 500 m in (a) autumn and (b) winter campaigns.

[Figure]

Figure III. Spatial distribution of the 15-year mean of PM$_{2.5}$ concentrations at a resolution of 1 km in the study region (left panel) and in the 4 major city clusters (right panel). Cited from Lin et al (2018).

New discussions are added in the revised manuscript (page 8, lines 19-28, page 9):

"**3.2.1 Vertical distribution of the major chemical components**

The profiles of the major PM$_{2.5}$ chemical components can generally be classified into three vertical gradients. The first category presents the highest concentration at ground level (type I). The second category shows the highest concentrations at 118 m (type II). And, the third category shows the highest concentration at 488 m (type III). The statistics of the three types in autumn and winter are listed in Table S1 and S2. We found that type II and type III were the major categories for sulfate, nitrate and

ammonium (SNA) in autumn, while those were most frequently observed in winter belong to type I and type II. Meanwhile, the OC and EC were most frequently seen in type I particles in both seasons.

Figure 3 shows the representative and average vertical profiles of $PM_{2.5}$, sulfate, ammonium, nitrate, OC, and EC mass concentration at the tower. In autumn, the vertical gradients for averaged sulfate, nitrate and ammonium were observed to be shallow, attributed to type II in which sulfate and nitrate concentrations were slightly higher at 118 m (Fig. 3a) while mean ammonium concentrations increased with height, a typical type III profile. Sulfate, nitrate, and ammonium concentrations on the polluted day (i.e., November 18, 2015) all increased with height, a typical type III profile. In particular, nitrate concentration was 1.5 times higher at 488 m than that at ground level, which will be further discussed in case studies. The vertical gradients for OC and EC were found to be much steeper than those for sulfate and ammonium, with the EC concentration 27.9% lower at 488 m than at ground level and OC concentration 34.0% lower at 488 m than at ground level (type I). The decrease in air pollutant concentrations with height is considered to be associated with ground-level sources (Zauli Sajani et al., 2018). No vertical gradients could be established for any of the measured PM components during clean days (e.g., as seen for October 31, 2015), which was likably attributed to the turbulent mixing of air pollutants within the boundary layer (Guinot et al., 2006).

In winter, averaged concentrations of sulfate and ammonium were generally observed to be higher at ground level than in the rest of their vertical gradients (type I) (Fig. 3b). However, concentrations of nitrate, OC and EC were higher at 118 m (type II). On clean days (i.e., Jan. 17, 2016) the vertical gradients for mean $PM_{2.5}$, SNA, OC, and EC mass concentrations were found to be shallow due probably to the well mixed air masses, while on polluted days (i.e., Jan. 2, 2016), the concentrations for sulfate, nitrate, ammonium and OC were higher at 118 m (type II). Our results showed that the vertical gradients for sulfate, nitrate and ammonium concentrations tend to be type II and type III in both autumn and winter seasons when the $PM_{2.5}$ concentrations were high (Table S1). The reasons were currently not clear, but they were probably due to local chemical

formation or regional transport of particles. However, back trajectory analysis of air masses showed that regional transport was unlikely the important source during the sampling time (Fig. S4) and then local chemical formation was likably the source that led to high SNA mass concentrations."

**Reference:**
Lin, C. Q., Liu, G., Lau, A. K. H., Li, Y., Li, C. C., Fung, J. C. H. and Lao, X. Q. High-resolution satellite remote sensing of provincial $PM_{2.5}$ trends in China from 2001 to 2015, Atmos. Environ., 180, 110-116, https://doi.org/10.1016/j.atmosenv.2018.02.045. 2018.

**[5]** I agree that using meteorology to interpret the aerosol data and to constrain the origin and formation of the different species is logical. But I don't agree that almost exclusively relying on WRF model output is reasonable. Why not at least validate those elements of the model predictions for which surface and/or satellite observations are available. Cloud cover and cloud base height are two that come to mind.

**[A]:** As suggested, we have now added the MODIS satellite images and ceilometer data in the supplementary of the revised manuscript. The ceilometer was mounted on the roof of South China Institute of Environment Sciences, Ministry of Ecology and Environment, which is about 4 km northeast of the Canton tower. However, we only obtained the winter pollution episode data because the ceilometer did not run during our autumn field study. MODIS satellite remote sensing images showed the cloud covers spreading over the PRD region (Figure S9 and Figure S10).

[Figure]

                        (b) November 19, 2015

Figure S9. MODIS images show the cloud covers over the PRD region during the autumn pollution episode (https://earthdata.nasa.gov/earth-observation-data/near-real-time/rapid-response)

[Figure]

(a) January 02, 2016                        (b) January 03, 2016

(c) Aerosol backscatter densities measured by ceilometer in Jan. 2 and Jan. 3, 2016.

**Figure S10.** Cloud cover from MODIS satellite (https://earthdata.nasa.gov/earth-observation-data/near-real-time/rapid-response) and cloud heights measured by ceilometer (Model CL-31, Vaisala Corp.) during the winter pollution episode.

We have modified and extended the section 3.3. More discussions have been

made in the revised manuscript (see in pages 14-16).

**[6]** For much of the discussion I believe it would be better to describe variations in absolute rather than relative concentrations. I recognize that for some explanations it is appropriate to describe differences in percent contribution of one or more species to the total concentration. But for other descriptions varying concentrations of other species unnecessarily complicates the results. One example is the conclusion on page 8, line 12 of favorable formation of the inorganic ion species based on relative changes in composition with height. The percentages would of course change in the same way if the concentrations of those species increased or those of other species decreased.

[A]: We appreciate the reviewer for providing valuable comments and suggestions. In fact, we did describe the variations of major $PM_{2.5}$ components using both absolute and relative concentrations. **In section 3.2.1**, we presented the vertical distribution of the major chemical components and discovered some vertical characteristics based on their **absolute** mass concentration profiles. In addition, we showed the **percentages** of different chemical species to fine and coarse particles at the three levels in **section 3.2.2**.

  As stated in the paper, we suggested that favorable formation of the inorganic ion species were in the higher levels based on relative changes in composition with height. Our results indicated that cloud processing and heterogeneous aqueous reactions together with unfavorable weather conditions were responsible for this phenomenon.

**Minor issues in the order in which they appear in the manuscript:**

**[1]** Page 3, Line 8: What are irregular sizes? And dust is usually used instead of sand.

**[A]:** We have modified in the text (page 3, line 13):

"Coarse-mode particles with large sizes and irregular shapes, such as dust particles, are usually produced from mechanical processes."

**[2]** Page 7, Line 24: SNA spelled out only later.

**[A]:** We changed in the text: "sulfate, nitrate, and ammonium (SNA)"

**[3]** Figure 2: Some explanation should be provided for the mismatch between the real-time and offline estimates of $PM_{2.5}$.

**[A]:** Thanks for the referee's suggestion. We added some explanations in the paper.

Page 7, lines 22-25: "The mismatch between the real-time $PM_{2.5}$ concentrations and the reconstructed $PM_{2.5}$ mass by combining the main components was likely due to sampling artefacts and lack of comprehensive offline $PM_{2.5}$ chemical analysis (Chow et al., 2015)."

Reference:

Chow, Judith C., Lowenthal, D. H., Chen, L. W. Antony, Wang, X. L. and Watson, J. G.: Mass reconstruction methods for $PM_{2.5}$: a review, Air Quality, Atmosphere & Health, 8(3), 243-263, https://doi.org/10.1007/s11869-015-0338-3, 2015.

**[4]** Figure 5: Some explanation should be provided for why the concentration of NH4 is highest at 488 m while that of $SO_4$ and $NO_3$ are not.

**[A]:** We added some explanations for this phenomenon.

Page 13, lines 6-10: "Figure 5 shows ammonium concentration was the highest in 488 m in autumn. The possible reason for this phenomenon might be that temperature (T) was lower and relative humidity (RH) was higher at 488 m, which was favorable for the partitioning of semi-volatile $NH_4NO_3$ into particle phase (Stelson and Seinfeld, 1982; Wang et al, 2012). This is supported by the evidence that nitrate concentration in fine particles generally increased with height (Figure 3)."

References:
Stelson, A.W., Seinfeld, J.H.: Relative humidity and temperature dependence of the ammonium nitrate dissociation constant. Atmos. Environ., 16(5), 983-992, https://doi.org/10.1016/0004-6981(82)90184-6, 2007.
Wang, X. F., Wang, W. X., Yang, L. X., Gao, X. M., Nie, W., Yu, Y. C., Xu, P. J., Zhou, Y., and Wang, Z.: The secondary formation of inorganic aerosols in the droplet mode through heterogeneous aqueous reactions under haze conditions, Atmos. Environ., 63, 68-76, http://dx.doi.org/10.1016/j.atmosenv.2012.09.029, 2012.

**[5]** Figures 5 and 6: I believe the 0.1 on all of the x-axes is supposed to be 0.01 and 0.1 is for some reason not shown.

**[A]:** We have modified these figures in the revised manuscript.

[Figure]

**Figure 5.** Mass concentration size distributions of the main chemical components measured at ground level, 118 m and 488 m in autumn. The dotted lines represent nonlinear fitting of the measured average size distribution. The error bars represent the sampling and analytical standard errors for each compound.

[Figure]

**Figure 6.** Mass concentration size distributions of the main chemical components measured at ground level, 118 m and 488 m in winter. The dotted lines represent nonlinear fitting of the measured average size distribution. The error bars represent the sampling and analytical standard errors for each compound.

---

## Editor Comment (EC1) · Barbara Ervens (Editor) · 20 Apr 2020

The review process of this manuscript was unusually long and involved comments of two referees during the discussion period and two more during the following revisions of the manuscript. All referees generally agreed that vertically-resolved aerosol mea-

surements on high towers are quite rare and may provide unique insights to sources and formation processes. However, the underlying processes, namely the formation of particulate sulfate, nitrate and ammonia, have been subject of many previous studies. Though the discussed data set is new, it is not sufficiently developed to substantially advance our understanding of these processes. Despite the major extension of the manuscript to include some model studies during the last phase of the review process, the advances and general implications for the scientific understanding of atmospheric chemistry and physics from this study are too limited to fulfill the standards for a research article in ACP. However, as it reports substantial new measurement results but with limited implications for atmospheric chemistry and physics, I decided to accept it for publication as a "Measurement Report" as it seems well suited for this manuscript category.

---

## Author Response (AR1)

Dear editor,

Thank you very much for your helps to handle this review. I am grateful to you and the reviewers for the valuable comments and suggestions provided. Our manuscript has been carefully revised based on the comments and suggestions. We respond to all comments of the reviewers. A revised manuscript is also prepared with the changes highlighted in red. Below, comments of the reviewers are given in normal font style and our responses are given in blue color. The changes to the manuscript are marked in red and also provided here.

Thank you again and looking forward to hearing from you soon.

Sincerely yours,

Shengzhen Zhou on behalf of the coauthors

**Response to the referee comments**

**Response to Anonymous Referee #1**

The study by Zhou et al. is an interesting one considering the fact that very high towers are not readily accessible to researchers. Having said that, similar studies have been performed around the world and despite claiming the uniqueness of the study it is not entirely unique, except for utilising perhaps the highest tower. However, it is not the height of the tower which makes any study unique, but instead scientific insights about the processes drawn from it. The study is not without significant drawbacks and needs significant improvement to warrant publication in the respected journal of Atmospheric Chemistry and Physics. Last but not least, English of the manuscript needs significant improvement as many sentences are unclear or dubious.

**[A]:** We thank the reviewer for valuable comments and suggestions and we have revised the paper accordingly. The revised manuscript has undergone a professional language editing and we sincerely hope that the English in the revised version could meet the ACP publishing standard.

In this study, size-segregated aerosol samples were concurrently collected at ground level, 118 m and 488 m of Canton tower in autumn and winter. Vertical mass size distributions of the PM chemical components were analyzed and the factors that affect their vertical variations were elucidated. The roles of in-cloud processes and heterogeneous aqueous reactions in haze formation were investigated in this subtropical urban area.

We have made thoroughly revision of introduction part to clarify our motivations and aims of the study. We have reanalyzed the vertical profiles of major PM components. Uncertainties and errors are included in this revised version. The haze formation schematic in the PRD region has been significantly improved and more discussions are added in this part.

Please kindly find our following point-by-point response. The reviewer's comments are in black and the authors' responses are in blue. Any changes made in the revision are highlighted in red.

**Major comments**

[1] The introduction needs significant improvement as overall interpretation of PM sources and processes is rather outdated, or straightforward or not consistent with the most recent fundamental papers. Very often the authors choose to reference either old papers missing out on recent ones, or choose to reference very recent, neglecting pioneering earlier papers. It is unclear what exactly were the goal and aims of the study other than utilising a very high tower. Those goals should normally arise from the earlier papers by identifying scientific knowledge gaps and which the authors choose to advance upon.

**[A]:** Thank you the reviewer for thoughtful comments. We have improved the introduction section as suggested. In addition, relevant and pioneering works are summarized in the revised manuscript. Major changes are made below. Please also see our responses to the minor comments [1]-[6].

A new paragraph was added in page 4, lines 14-28 and page 5, lines 1-3: "Severe aerosol pollutions frequently occur in China, as exemplified by three cities groups in the Jing-Jin-Ji (Beijing, Tianjin, and Hebei province), the Yangtze River Delta, and the Pearl River Delta regions. State-of-the-art air quality models still often fail to simulate the observed high PM2.5 concentrations even after including aerosol-radiation-meteorology feedback, indicating that key atmospheric chemical processes, such as heterogeneous and multiphase reactions, are lacking in models for secondary aerosol formation (Zheng et al, 2015; Cheng et al, 2016). To improve the understanding of haze formation, models will require updated kinetic and mechanistic data of multiphase chemistry and quantification of the aerosol formation through heterogeneous reactions under real atmospheric conditions (Zheng et al, 2015; An et al, 2019). Additionally, more consistent evidences of aerosol formation through heterogeneous reactions are needed from field measurements, laboratory experiments and model simulations. Field studies showed that extremely high PM2.5 concentrations usually occurred under high relative humidity conditions (Sun et al, 2014; Wang et al, 2014). Heterogeneous aqueous phase reactions in the cloud liquid water and in aerosol water can promote secondary aerosol formation (Seinfeld et al, 2006; Ervens et al, 2015; McNeill, 2015; Cheng et al, 2016). It is hence critical to investigate the aerosol sources and formation mechanisms by measuring size-resolved PM components vertically using a tall tower, where they can be strongly influenced by the dynamic variations of atmospheric boundary layer and cloud processing."

In addition, we have clarified the objectives of this study in the revised manuscript.

Page 5, lines 13-17: "The objectives of this study are to (1) analyze the vertical mass size distribution of the PM chemical components and the factors that affect their vertical variations; and (2) investigate the roles of in-cloud processes and heterogeneous aqueous reactions in secondary aerosol formation and the implication for haze pollution in subtropical urban areas."

[2] The paper currently stands more like a report rather than a scientific paper. It presents data, but lacks coherent view. More often than not the authors seek consistency with other studies or providing references which support or fit their data. Taken altogether, the paper is currently a collection of interpretations which are not always consistent with each other and most importantly lacking conclusive findings which would advance rather than confirm already known processes or phenomena.

[A]: We thank the reviewer for the valuable comments on the presentation of the paper. In this paper, we analyze the vertical mass size distribution of the PM chemical components and the factors that affect their vertical variations. In addition, we also investigate the roles of in-cloud processes and heterogeneous aqueous reactions in haze formation in the subtropics urban areas.

We have carefully addressed the reviewer's concerns and made a thorough revision in the introduction section. In addition, we have added several discussions about the haze formation mechanisms in section 3.3.

Please see more details in our point-by-point response to your specific comments below. We believe that revised manuscript has been improved in this regard.

[3] The study is lacking an overview of all the profiles, splitting into certain categories and introducing the scope and variability of the data set first. There is a complete lack of uncertainty and error analysis. Analytical and sampling uncertainty would propagate into vertical profile uncertainties which would then make profiles or concentrations at different heights significantly different or not. The authors choose to select specific episodes or profiles which are most obvious or interpretable and neglecting which are not. Selected profiles are certainly insightful, but only when put into overall context.

[A]: We thank the reviewer for the valuable comments. We have added two tables (Tables S1 and S2) in the supplementary to show an overview of all the profiles. Uncertainties and errors were included in Fig. 3.

---

## Editor Decision (ED1)

**Major comments**

1) The formation processes of sulfate and nitrate are well known. Thus, the current study does not contribute much to the understanding of the underlying processes that lead to aerosol formation.

In order to make the current study a substantial contribution to our understanding of general implications for atmospheric science rather than being focused on the investigation of local air pollution, the manuscript's focus needs to be expanded.

Referee #4 suggests to apply WRF-Chem not only to predict the meteorological parameters but also to simulate formation of sulfate and nitrate for comparison to measurements. This would provide useful information about model performance and it would broaden the scope of the manuscript as general conclusions on model performance could be drawn using your unique data set of size-segregated vertical aerosol profiles.

2) The formation processes and sources of organic aerosol are much more uncertain than those of sulfate and nitrate. Given that a large mass fraction of the aerosol in the study region is composed of organics, some discussion should be dedicated to them. For example, Referee #4 points out that the different vertical gradients of organics compared to inorganics may give evidence of different sources. This idea should be elaborated on and the discussion extended accordingly.

3) Referee #3 suggests improving the structure of your discussion and to more clearly differentiating between hypotheses and clear evidence of aerosol processing based on your observations. Your statements should be stronger and more specific; at many places they seem vague, e.g.

p. 1, l. 10: The results from pollution case studies further showed that atmospheric aqueous-phase and 10 heterogeneous reactions together with adverse weather conditions, such as temperature inversion and calm wind, resulted in the autumn and winter haze pollution in the PRD region

p. 9, l. 21: However, back trajectory analysis of air masses showed that regional transport was unlikely the important source during the sampling time (Fig. S4) and then local chemical formation was likably the source that led to high SNA mass concentrations.

p. 10, l. 6: indicating the favorable secondary formation or regional transport of aerosols at the higher altitude

p. 17, l. 9: …. suggesting different nitrate formation mechanisms

**Minor comments**

- Referee #1 appreciates the detailed uncertainty analysis that you provided in the response to the referee report. It should be included in the manuscript.

- Both Referee #2 and #3 pointed out that the hygroscopic growth of particles might lead to a significant shift in diameters which might affect conclusions about aerosol size distributions at different heights. Please take into account the numbers for growth factors etc as suggested by Referee #3 and discuss possible implications for your conclusions on aerosol mass formation vs aerosol growth by water uptake.

- At several places in the manuscript, you use 'heterogeneous aqueous phase reaction' or seem to use 'heterogeneous' and 'multiphase' reactions equivalently. Please check carefully for the correct use of terminology: 'Heterogeneous processes' are processes that occur on surfaces (droplet, particles) where the

reactants are in two different phases (gas and condensed phase). 'Multiphase processes' are processes that occur in the bulk of the aqueous phase into which the reactants might be taken up from or products released to the gas phase (Ravishankara, A. R.: Heterogeneous and Multiphase Chemistry in the Troposphere, Science, 276, 1058–1065, 1997.)

- Numbers should be rounded to significant digits throughout the manuscript, e.g. $44 \pm 14$ instead of $44.1 \pm 14.9$

- p. 8, l.11: What are abnormal days, as opposed to 'normal days'?

- Nitrate formation from hydrolysis of $N_2O_5$ is well known. In the atmosphere, likely no completely dry particles exist as mixed particles have a very low efflorescence relative humidity and continuously undergo efflorescence/deliquescence cycles. The discussion on op. 12 can be shortened.

- p. 14, l. 10-14: These lines seem out of place in the 'OC and EC' section.

- p. 15, l. 10-12: Was the increase in $SO_2$ sufficient to lead to the observed decrease in SOR? In other words, does this imply that no sulfate formation occurred?

---

## Author Response (AR2)

**Responses to comments**

Dear editor,

We thank the editor and reviewers for their valuable comments and suggestions. We have addressed all raised issues in the revision accordingly. Please kindly find our following point-by-point response. The reviewer's comments are in black and the authors' responses are in blue. Changes that have been made in the revised manuscript are highlighted in red.

**Major comments**

**1)** The formation processes of sulfate and nitrate are well known. Thus, the current study does not contribute much to the understanding of the underlying processes that lead to aerosol formation.

In order to make the current study a substantial contribution to our understanding of general implications for atmospheric science rather than being focused on the investigation of local air pollution, the manuscript's focus needs to be expanded.

Referee #4 suggests to apply WRF-Chem not only to predict the meteorological parameters but also to simulate formation of sulfate and nitrate for comparison to measurements. This would provide useful information about model performance and it would broaden the scope of the manuscript as general conclusions on model performance could be drawn using your unique data set of size-segregated vertical aerosol profiles.

**[A]:** We thank the editor and reviewers for their valuable comments and suggestions.

Vertical profiles of size-segregated aerosol mass and chemical compositions are crucial for elucidating aerosol sources and formation as well as evaluating the performance of atmospheric models. Such data have, however, still scarcely been reported due to the limitations of observational methods and platforms. We agree with the editor and the reviewer that we should provide information about the current model performance by comparison with our measurement data. As suggested, we simulated the vertical concentration profiles of sulfate and nitrate using WRF-Chem model (Weather Research and Forecasting Model coupled with online chemistry model in version 3.7.1).

The results showed that sulfate was generally underestimated in WRF-Chem model at the upper level, while was in relatively good agreement with observation at the surface. Possible reasons for the underestimations of sulfate are: (1) $SO_2$ precursors were underestimated at the upper levels (by about 45% to 77%, table S6), possibly due to the insufficient upward transport of $SO_2$ in the current model, especially in urban area where the urban canopy is low in resolution; (2) heterogeneous/multiphase formations of sulfate in droplets or aerosol water have not been considered enough in current model (Chen et al., 2016; Cheng et al. 2016).

The WRF-Chem model overestimated nitrate concentrations in both seasons. We put forward three potential explanations: (1) underestimation of $SO_4^{2-}$, which consumes less $NH_3$, facilitates $NH_3NO_4$ formation in the fine model at the high levels (Tuccella et al., 2012); (2) heterogeneous reaction of $HNO_3$ on coarse-mode sea salt

aerosols, however, will reduce the formation of fine-mode nitrate (Chen et al, 2016). Therefore, sea salt emissions in current model should be evaluated especially over the coastal regions; (3) the cloud fraction and liquid water content may not be well simulated in the model. We added above discussion in the revision.

The model performance and possible implications have now been added in a new section (section 3.4, page 17, lines 20-28 and page 18, lines 1-22) and supplementary.

**"3.4 Model simulation and implications**

Vertical concentration distributions of sulfate, nitrate and ammonium were further simulated by WRF-Chem model. The description and configuration of the model can be found in the Supplementary. Figures S13 and S14 show the simulated vertical concentration profiles of sulfate, nitrate and ammonium in autumn and winter and their comparisons with observation. Sulfate was generally underestimated in WRF-Chem model at the upper level, while was relatively in good agreement with observation at the surface. Possible reasons for the underestimations of sulfate are: (1) $SO_2$ precursors were underestimated at the upper levels (by about 45% to 77%, table S6), possibly due to the insufficient upward transport of $SO_2$ in the current model, especially in urban area where the urban canopy is low in resolution; (2) heterogeneous/multiphase formations of sulfate in droplets or aerosol water have not been considered enough in current model (Chen et al., 2016; Cheng et al. 2016). Nitrate was overestimated by WRF-Chem model. Here three reasons were put forward: (1) the underestimation of $SO_4^{2-}$ at the upper levels, which consumes less $NH_3$, facilitates the formation of $NH_3NO_4$ formation in the fine-mode (Tuccella et al., 2012); (2) heterogeneous reaction of $HNO_3$ on coarse-mode sea salt aerosols, however, will reduce the formation of fine-mode nitrate (Chen et al, 2016). Therefore, sea salt emissions in current model should be evaluated especially over the coastal regions; (3) the cloud fraction and liquid water content may not be well simulated in the model. For ammonium, the simulated concentrations were overall consistent with the measured ones except for being slightly overestimated at ground level. The large discrepancies between observation and simulation on sulfate and nitrate suggested that physical and chemical mechanisms in current WRF-Chem model still need to be improved to better predict aerosol mass and composition. Based on our observation, in-cloud aqueous phase reactions and heterogeneous reactions should play important roles in sulfate and nitrate formation, which need to be refined in the model. Evaluation of WRF-Chem model incorporating the above-mentioned mechanisms is beyond the scope of this study and in-depth investigation needs to be done in future. Hence, more studies, such as long-term aerosols and high frequency micrometeorological measurements (Valiulis et al., 2002; Ceburnis et al., 2008; Ervens, 2015), are needed to identify the key aerosol sources and formation pathways, and to further improve the air quality models."

[Figure]

Figure S13. The vertical concentration profiles of sulfate, nitrate, and ammonium in $PM_{2.5}$ during autumn (The red solid lines are the average modeled concentrations and the shaded regions indicate the minimum and maximum values of the simulation; the average measurement data were in black with horizontal error bars).

[Figure]

Figure S14. The vertical concentration profiles of sulfate, nitrate, and ammonium in $PM_{2.5}$ during winter (The red solid lines are the average modeled concentrations and the shaded regions indicate the minimum and maximum values of the simulation; the average measurement data were in black with horizontal error bars).

References:

Ceburnis, D., Dowd, C. D. O', Jennings, G. S., Facchini, M. C., Emblico, L., Decesari, S., Fuzzi, S., and Sakalys, J.: Marine aerosol chemistry gradients: Elucidating primary and secondary processes and fluxes, Geophy. Res. Let., 35(7), L07804,

https://doi.org/10.1029/2008GL033462, 2008.

Chen, D., Liu, Z., Fast, J. and Ban, J.: Simulations of sulfate-nitrate-ammonium (SNA) aerosols during the extreme haze events over northern China in October 2014, Atmos. Chem. Phys., 16(16), 10707–10724, doi:10.5194/acp-16-10707-2016, 2016.

Chen, Y., Cheng, Y., Ma, N., Wolke, R., Nordmann, S., Schüttauf, S., Ran, L., Wehner, B., Birmili, W., van der Gon, H. A. C. D., Mu, Q., Barthel, S., Spindler, G., Stieger, B., Müller, K., Zheng, G. J., Pöschl, U., Su, H., and Wiedensohler, A.: Sea salt emission, transport and influence on size-segregated nitrate simulation: a case study in northwestern Europe by WRF-Chem, Atmos. Chem. Phys., 16, 12081-12097, 10.5194/acp-16-12081-2016, 2016.

Cheng, Y. F., Zheng, G. J., Wei, C., Mu, Q., Zheng, B., Wang, Z. B., Gao, M., Zhang, Q., He, K. B., Carmichael, G., Pöschl, U., and Su, H.: Reactive nitrogen chemistry in aerosol water as a source of sulfate during haze events in China, Sci. Adv., 2, e1601530, doi: 10.1126/sciadv.1601530, 2016.

Ervens, B.: Modeling the Processing of Aerosol and Trace Gases in Clouds and Fogs. Chem. Rev., 115(10), 4157-4198, http://dx.doi.org/10.1021/cr5005887, 2015.

Tuccella, P., Curci, G., Visconti, G., Bessagnet, B., Menut, L., and Park, R. J.: Modeling of gas and aerosol with WRF/Chem over Europe: Evaluation and sensitivity study, Journal of Geophysical Research: Atmospheres, 117, 10.1029/2011jd016302, 2012.

Valiulis, D., Ceburnis, D., Sakalys, J., and Kvietkus, K.: Estimation of atmospheric trace metal emissions in Vilnius City, Lithuania, using vertical concentration gradient and road tunnel measurement data, Atmos. Environ., 36(39-40), 6001-6014, https://doi.org/10.1016/S1352-2310(02)00764-1, 2002.

**Supplementary materials**

**"2.1 Description and configurations of WRF–Chem model**

The model used in this study is the chemistry version of the WRF model (WRF–Chem v3.7.1). The WRF model is a mesoscale non-hydrostatic meteorological model that includes several options for physical parameterizations of the planetary boundary layer (PBL), cloud processes and land surface (Skamarock et al., 2008). The chemistry version is a version of WRF coupled with an "online" chemistry model, in which meteorological and chemical components of the model are predicted simultaneously (Grell et al., 2005; Fast et al., 2006). Emission, chemical formation and removal, transport and deposition are considered when WRF–Chem predicts the chemical components. In the numerical modeling, a triple-nested grid with 27/9/3 km resolution domain and 39 layers in vertical is set (Figure S12). The chemistry only run in domain d03 (3 km resolution) and MOZART boundary condition is used for the chemistry run. The simulation period is conducted from 0000 UTC 18 October to 0000 UTC 28 November 2015 as the autumn time period and 0000 UTC 25 December 2015 to 0000 UTC 28 January 2016 as the winter time period. The National Center for Environmental Protection (NCEP) $1° \times 1°$ FNL (Final operational global reanalysis) data are applied as the initial and boundary condition in WRF modeling. The physical and chemical parameterization configurations for WRF-Chem model can be found in Table S4. A region local anthropogenic emission inventory is used within the PRD region (Zheng et al., 2009), which is updated to the year of 2015.

For other areas, the Multi-resolution Emission Inventory for China (MEIC, http://meicmodel.org/) is used in this study. MEGAN model version 2.1 is also used for providing the biogenic emissions (Guenther et al., 2006).

[Figure]

**Figure S12.** The simulation domain of WRF-Chem model

**Table S4.** Physical and chemical parameterization configurations for WRF-Chem model

| Process | Option |
|---|---|
| Microphysics scheme | Morrison (2 moments) |
| Cumulus scheme | Kain-Fristsch |
| Longwave radiation scheme | RRTM |
| Shortwave radiation scheme | Dudhia |
| Boundary-layer scheme | YSU |
| Land-surface scheme | unified Noah |
| Urban Surface scheme | UCM |
| Gas-phase mechanism scheme | CBM-Z |
| Photolysis scheme | Fast-J |
| Aerosol scheme | MOSAIC |

**2)** The formation processes and sources of organic aerosol are much more uncertain than those of sulfate and nitrate. Given that a large mass fraction of the aerosol in the study region is composed of organics, some discussion should be dedicated to them. For example, Referee #4 points out that the different vertical gradients of organics compared to inorganics may give evidence of different sources. This idea should be elaborated on and the discussion extended accordingly.

***[A]:*** We agree that the formation processes and sources of organic aerosol are much more uncertain than those of sulfate and nitrate. Current atmospheric models are not able to accurately predict the amount of organic aerosols formed due to these high uncertainties.

The vertical distributions of OC and EC in $PM_{2.5}$ have been discussed and compared with inorganic species in the revision (section 3.2.1, page 9, lines 15-19 and page 10):

[revised manuscript text omitted]

**3)** Referee #3 suggests improving the structure of your discussion and to more clearly differentiating between hypotheses and clear evidence of aerosol processing based on your observations.

*[A]***:** We have improved the structure of our discussion in revision.

Based on the vertical mass size distribution study, our hypotheses are that in-cloud aqueous reactions and heterogeneous reactions were the important aerosol formation pathways in this coastal urban area under high relative humidity. Then, we carried out aerosol pollution case studies to prove our hypotheses. To make the discussion clear, we have added some sentences in section 3.3:

"In the previous sections, our results revealed that in-cloud aqueous reactions and heterogeneous reactions were important aerosol formation pathways in this coastal urban area under high relative humidity. Here we consider two PM pollution episodes (E1 and E2) to investigate these mechanisms for haze formation in autumn and in winter (Fig. 2)."

We further simulated the vertical concentration profiles of sulfate, nitrate and ammonium using WRF-Chem model, which are compared with the measured values. The model performance was also evaluated. This information was added in a new paragraph (section 3.4).

Your statements should be stronger and more specific; at many places they seem vague, e.g.

*[A]*: We thank the reviewer for the valuable comments and we have now modified those sentences below for clarification.

p. 1, l. 10: The results from pollution case studies further showed that atmospheric aqueous-phase and 10 heterogeneous reactions together with adverse weather conditions, such as temperature inversion and calm wind, resulted in the autumn and winter haze pollution in the PRD region.

*[A]*: Page 2, lines 10-13: "Case studies further showed that atmospheric aqueous-phase and heterogeneous reactions could be important mechanisms for sulfate and nitrate formation, which, in combination with adverse weather conditions, such as temperature inversion and calm wind, led to haze formation during autumn and winter in the PRD region."

p. 9, l. 21: However, back trajectory analysis of air masses showed that regional transport was unlikely the important source during the sampling time (Fig. S4) and then local chemical formation was likably the source that led to high SNA mass concentrations.

*[A]*: Page 10, lines 1-3: "However, back trajectory analysis showed that local chemical formation contributed to high SNA mass concentrations rather than regional transport during the sampling time (Fig. S4)."

p. 10, l. 6: indicating the favorable secondary formation or regional transport of aerosols at the higher altitude.

*[A]*: Page 10, line 16: "indicating more favorable secondary formation and aging processes of aerosols at higher altitudes."

p. 17, l. 9: …. suggesting different nitrate formation mechanisms

*[A]*: Page 19, lines 5-7: "Bi-modal size distributions in autumn and a unimodal mode in winter were observed for nitrate, suggesting different formation pathways for nitrate in different seasons."

**Minor comments**

**[1]** - Referee #1 appreciates the detailed uncertainty analysis that you provided in the response to the referee report. It should be included in the manuscript.

*[A]:* We have added the detailed uncertainty analysis in the text and supplementary.

Page 6, lines 8-15: The three impactors (or samplers) were calibrated using mass flow meter (TSI, model 4040) in laboratory prior to the measurements. The flow rates of the impactors were measured at the beginning of the sampling. At the end of the sampling, the flow rates were recorded again. If the flow rate of each impactor at the beginning and end of the sampling differed by more than 10%, the sample was marked as suspect and the data was discarded. The average flow rates at the beginning and end of the sampling was used to be the sampling flow rate. In addition, a magnehelic pressure gauge was used to monitor the inlet flow rate through the impactor. The pressure drop was also recorded at the beginning and end of the sampling.

Page 6, lines 22-25: We estimated the impacts of temperature and pressure on the flow rate due to the sampling heights which are less than 5%. (refer to supplementary).

**[2]** - Both Referee #2 and #3 pointed out that the hygroscopic growth of particles might lead to a significant shift in diameters which might affect conclusions about aerosol size distributions at different heights. Please take into account the numbers for growth factors etc as suggested by Referee #3 and discuss possible implications for your conclusions on aerosol mass formation vs aerosol growth by water uptake.

*[A]*: We thank the reviewer for the valuable comments. We agree that the shifts of the particles sizes would happen due to increase of relative humidity. Therefore, we estimated the RH effects on the shift in the size cut-off threshold induced by hygroscopic growth.

In this study, the monthly averaged relative humidity was around 80% in both seasons. We assume that the hygroscopic growth factor, independence of particle size, is about 1.4 and 1.5 respectively for ammonium sulfate and ammonium nitrate at 80% RH (Tang, 1996). A maximum overestimated value of about 10%, 13% and 6% for droplet mode sulfate was obtained at ground level, 118 m and 488 m, respectively. The droplet mode nitrates were overestimated to be about 11%, 16%, and 6% at ground level, 118 m and 488 m, respectively.

Chen et al (2018) explored the influence of cut-off shift on filter-based particle sampling driven by hygroscopic growth. They concluded that the influence is generally negligible (less than 7%) in urban areas, but need to be considered (about 10–20%) in continental background areas in Europe. They also recommended that this influence needs to be assessed individually for each measurement period even at the same location, since it is highly dependent on ambient conditions.

Based on the evaluation, we think the influence of RH on the size shift indeed exists, but is unlikely to change the main conclusion of this paper. Sulfate and nitrate concentrations are still mainly dominant in droplet mode. We have now added the following paragraph as a caveat in the revision.

Page 11, line 22-: The air was not dried upstream of the impactor and therefore relative humidity would influence the size distributions of different chemical species. In this study, the monthly averaged relative humidity was around 80% in both seasons.

We assume that the hygroscopic growth factor is independent of particle size and is about 1.5 for both ammonium sulfate and ammonium nitrate at 80% RH (Tang, 1996). A maximum overestimated value of about 10%, 13% and 6% for droplet mode sulfate was obtained at ground level, 118 m and 488 m, respectively. The droplet mode nitrates were overestimated to be about 11%, 16%, and 6% at ground level, 118 m and 488 m, respectively. Our results are similar to those in Chen el al (2018), which concluded that the influence of cut-off shift on filter-based particle sampling driven by hygroscopic growth is generally negligible (less than 7%) in urban areas, but need to be considered (about 10–20%) in continental background areas in Europe. The influence from relative humidity on the size distribution is indeed present based on the average particle concentrations in the droplet mode in this study; however, it is unlikely to change our conclusion that in-cloud processing contributed to the droplet mode aerosols (Table S3).

References:

Chen, Y., Wild, O., Wang, Y., Ran, L., Teich, M., Größ, J., Wang, L., Spindler, G., Herrmann, H., van Pinxteren, D., McFiggans, G., and Wiedensohler, A.: The influence of impactor size cut-off shift caused by hygroscopic growth on particulate matter loading and composition measurements, Atmospheric Environment, 195, 141-148, https://doi.org/10.1016/j.atmosenv.2018.09.049, 2018.

Tang, I. N.: Chemical and size effects of hygroscopic aerosols on light scattering coefficients, Journal of Geophysical Research: Atmospheres, 101, 19245-19250, 10.1029/96jd03003, 1996.

**[3]** - At several places in the manuscript, you use 'heterogeneous aqueous phase reaction' or seem to use 'heterogeneous' and 'multiphase' reactions equivalently. Please check carefully for the correct use of terminology: 'Heterogeneous processes' are processes that occur on surfaces (droplet, particles) where the reactants are in two different phases (gas and condensed phase). 'Multiphase processes' are processes that occur in the bulk of the aqueous phase into which the reactants might be taken up from or products released to the gas phase (Ravishankara, A. R.: Heterogeneous and Multiphase Chemistry in the Troposphere, Science, 276, 1058–1065, 1997.)

*[A]:* We thank the reviewer for pointing this out. We have double checked and modified accordingly in the revision.

Page 3, line 13: "heterogeneous aqueous reactions" should be "multiphase reactions".
Page 5, line 3: "Heterogeneous aqueous phase reactions" should be "Multiphase reactions".
Page 5, line 19: "heterogeneous aqueous reactions" should be "multiphase reactions".
Page 17, line 4: "heterogeneous aqueous reactions" changed to "aqueous reactions".

**[4]** - Numbers should be rounded to significant digits throughout the manuscript, e.g. 44 ±14 instead of 44.1 ±14.9

*[A]:* All the numbers have been checked and rounded according to the reviewer's

suggestion.

**[5]** - p. 8, l.11: What are abnormal days, as opposed to 'normal days'?
*[A]:* As suggested by the reviewer, we changed "normal days" to "typical days".

**[6]** - Nitrate formation from hydrolysis of $N_2O_5$ is well known. In the atmosphere, likely no completely dry particles exist as mixed particles have a very low efflorescence relative humidity and continuously undergo efflorescence/deliquescence cycles. The discussion on op. 12 can be shortened.
*[A]:* We have deleted those sentences associated with hydrolysis of $N_2O_5$.

**[7]** - p. 14, l. 10-14: These lines seem out of place in the 'OC and EC' section.
*[A]:* We rephrased this sentence in the revision. (Page 15, lines 2-9):
"EC showed a general bimodal peak in winter, with a broad EC peak in accumulation mode and a sharp peak at around 1.0–1.44 μm, with the highest concentration in this size range 1.0-1.44 at 188 m, suggesting that its source might be different from the other two levels. One possible reason for the abnormally high concentrations of EC was the influence of local point sources (i.e., high chimneys from power plants and industry) around Guangzhou which emitted elevated concentrations of air pollutants from combustion sources. We also found that the concentrations of co-emitted $SO_2$ and CO were the highest at 188 m among the three heights (Fig. S7)."

**[8]** - p. 15, l. 10-12: Was the increase in $SO_2$ sufficient to lead to the observed decrease in SOR? In other words, does this imply that no sulfate formation occurred?
*[A]:* We thank the reviewer for pointing this out. It was likely in our study that ambient air at higher elevations was impacted more by emissions from nearby combustion sources, where the air masses were fresh. Therefore, we observed lower SOR (SOR = $n-SO_4^{2-}$ / ($n-SO_4^{2-}$ + $n-SO_2$) at higher altitudes. That does not mean, however, that no sulfate formation occurred but the new released $SO_2$ will compensate for part of the sulfate converted from $SO_2$ at high altitudes, leading to decrease of the SOR.
We have added this discussion in the text (page 16, line 7- 10):
"The concentration of $SO_2$ increased with height (12.4, 16.1, 27.0 μg m$^{-3}$ at ground, 118 m, and 488 m, respectively), suggesting that it was impacted more by emissions from local sources, where the air masses were fresh. The newly emitted $SO_2$ at high altitudes compensated for part of the sulfate converted from $SO_2$, leading to decrease of SOR with heights."

**Response to Anonymous Referee #1**

The paper improved dramatically and is now suitable for publication. The authors did a great job and their efforts commendable.

**Few minor technical comments:**
[1]-Use "typical (days", instead of "normal".
*[A]:* we changed "normal days" to "typical days".

[2]-I wonder why uncertainty analysis which was well explained to the reviewers did not make into the paper? It can be separated into dedicated chapter.
*[A]:* We added the detailed uncertainty analysis in the text.
Page 6, lines 8-15: The three impactors (or samplers) were calibrated using mass flow meter (TSI, model 4040) in laboratory prior to the measurements. The flow rates of the impactors were measured at the beginning of the sampling. At the end of the sampling, the flow rates were recorded again. If the flow rate of each impactor at the beginning and end of the sampling differed by more than 10%, the sample was marked as suspect and the data was discarded. The average flow rates at the beginning and end of the sampling was used to be the sampling flow rate. In addition, a magnehelic pressure gauge was used to monitor the inlet flow rate through the impactor. The pressure drop was also recorded at the beginning and end of the sampling.

Page 6, lines 22-25: We estimated the impacts of temperature and pressure on the flow rate due to the sampling heights. The results showed that impacts of pressure and temperature on the flow rate are less than 5% (Refer to supplementary).

**Response to Anonymous Referee #3**

The Authors present original aerosol measurements taken on a tall city tower, including rare observations of size-segregated chemical composition. Measurement periods are short, and the Authors focused on the analysis of specific events. Their results are interesting because they were able to show that aerosol chemical components can be spatially segregated not only horizontally (because of the presence of the buildings and of the heterogeneity of the sources) but also vertically. The analysis of air mass transport and thermodynamics in the lower few hundreds meters above the ground can be blamed for such variability. In spite of such complexity, the Authors' conclusions in the abstract seem very shallow. Just a generic mention to the role of heterogeneous chemistry is given. The quality of presentation is poor, and I struggled to understand what specific day of the campaign some of the figures referred to (e.g. Figure 8) and also why that day and not others? The Authors claim that the presence of low-level clouds was crucial for the production of secondary aerosols on certain days. But they do not discuss the presence/absence of clouds on other days, when concentration vertical profiles are more flat. The full elaboration lacks of a coherent analysis, and sometimes hypotheses and conclusions (e.g., about the importance of aqueous phase chemistry) are mixed up in the discussion.

*[A]*: We thank the reviewer for the comments. In this study, size-segregated aerosol samples were concurrently collected at ground level, 118 m and 488 m of Canton tower in autumn and winter. Vertical mass size distributions of the PM chemical components were analyzed and the factors that affect their vertical variations were elucidated. The roles of in-cloud aqueous reactions and heterogeneous reactions in haze formation were investigated in this subtropical urban area. Moreover, vertical distributions of sulfate, nitrate and ammonium were simulated by the WRF-Chem model. The large discrepancies between observation and simulation on sulfate and nitrate suggested that physical and chemical mechanisms in the current model still need to be improved to better predict the aerosols masses and compositions.

In order to make the paper clear and more readable, we adjusted the structure, and a new paragraph on model simulation and implications was added in the revised manuscript.

**Specific comments:**
The Referee#2's comment on the effect of hygroscopic growth on measured impactor size-distribution, in my view, was only partially addressed by the Authors. In the study of Meng and Seinfeld (1993) quoted by the Authors, the ratio between the diameters of the droplet mode and the condensation mode was 3.5 (= 0.7 vs. 0.2), while here such ratio is only 2.3 for sulphate (= 0.75 vs 0.33; section 1 in the Supplementary). Considering that the hygroscopic growth factor for ammonium sulphate at RH 80% is about 1.4 (and obviously greater at RH values approaching the saturation), I would be more cautious in concluding that ambient RH was irrelevant in contributing to shaping the measured size-distributions.

*[A]*: We thank the reviewer for the valuable comments. We agree that the shifts of the

particles sizes would happen due to the increase of relative humidity. Therefore, we estimated the RH effects on the shift in the size cut-off threshold induced by hygroscopic growth.

In this study, the monthly averaged relative humidity was around 80% in both seasons. We assume that the hygroscopic growth factor, independence of particle size, is about 1.4 and 1.5 respectively for ammonium sulphate and ammonium nitrate at 80% RH (Tang, 1996). A maximum overestimated value of about 10%, 13% and 6% for droplet mode sulphate was obtained at ground level, 118 m and 488 m, respectively. The droplet mode nitrates were overestimated to be about 11%, 16%, and 6% at ground level, 118 m and 488 m, respectively.

Chen et al (2018) explored the influence of cut-off shift on filter-based particle sampling driven by hygroscopic growth. They concluded that the influence is generally negligible (less than 7%) in urban areas, but need to be considered (about 10–20%) in continental background areas in Europe. They also recommended that this influence needs to be assessed individually for each measurement period even at the same location, since it is highly dependent on ambient conditions.

Based on the evaluation, we think the influence of RH on the size shift indeed exists, but is unlikely to change the main conclusion of this paper. Sulfate and nitrate concentrations are still mainly dominant in droplet mode. We have now added the following paragraph as a caveat in the revision.

Page 11, line 25-: "The air was not dried upstream of the impactor and therefore relative humidity would influence the size distributions of different chemical species. In this study, the monthly averaged relative humidity was around 80% in both seasons. We assume that the hygroscopic growth factor is independent of particle size and is about 1.5 for ammonium sulfate and ammonium nitrate at 80% RH (Tang, 1996). A maximum overestimated value of about 10%, 13% and 6% for droplet mode sulfate was obtained at ground level, 118 m and 488 m, respectively. The droplet mode nitrates were overestimated to be about 11%, 16%, and 6% at ground level, 118 m and 488 m, respectively. Our results are similar to those in Chen el al (2018), which concluded that the influence of cut-off shift on filter-based particle sampling driven by hygroscopic growth is generally negligible (less than 7%) in urban areas, but need to be considered (about 10–20%) in continental background areas in Europe. The influence from relative humidity on the size distribution is indeed present based on the average particle concentrations in the droplet mode in this study; however, it is unlikely to change our conclusion that in-cloud processing contributed to the droplet mode aerosols (Table S3)."

**Response to Anonymous Referee #4**

This paper discusses size segregated measurements of aerosol particle chemistry (major ions, OC and EC) conducted at three vertical levels (ground, 118 m-agl and 488 m-agl) in an urban area of China. In addition to trajectories, the WRF model is used as a representation of low-level vertical profiles of temperature, RH and winds in the area of the measurements.

Overall, I find the presentation and writing of the current version to be clear.

The emphasis of the work appears to be to demonstrate that aqueous-phase production of sulphate plays a significant role in the development of the local aerosol and its vertical gradient. Additionally, it is suggested that nitrate formation is influenced by aqueous-phase processes and the presence of sea salt. In my opinion, the authors have presented fairly convincing arguments that these processes are likely involved in a potentially significant way. However, all of these processes have been known and studied for many years, and, as I understand it, the issue of how they affect local observations is really not in the purview of ACP. Therefore, as the paper now stands, I do not believe it is suitable for publication in ACP.

I have a few suggestions: 1) I think it would be valuable, and something that might have more universal appeal, to test WRF-CHEM against your observations to see how well the model does at reproducing your results. To do this, some local observations of state parameters and winds might be needed. 2) The carbonaceous components of the measurements are really under-emphasized in the paper. More emphasis on the size distributions of EC and OC in a relatively high emissions area, coupled with comparisons for other urban areas, regional and remote results might help. 3) In its current form, the paper might be more suitable for Atmospheric Environment, which has more of a focus on urban issues.

*[A]*: We sincerely thank the reviewer for the valuable comments, which were of great help in revising the manuscript.

In the present study, vertical concentration profiles of aerosol components in different size ranges were studied based on a ~ 610 m tall tower. The vertical concentration profiles of sulfate, nitrate and ammonium and their mass size distribution characteristics suggested that cloud processing and heterogeneous reaction could be important formation pathways for them, which should largely contribute to the aerosols pollution in this coastal urban region. The characteristics of carbonaceous aerosols were also discussed to provide more information on the aerosol sources. Further, we carried out aerosols pollution case studies to prove that cloud processing and heterogeneous reactions played important roles in the aerosols formation and haze pollutions in this area. Although we have already well known about the formation processes of sulfate and nitrate, the dominant formation pathways, and their contribution to haze pollution and climatic effects in different regions may be different and should be considered more in the current models.

As suggested, we simulated the vertical concentration profiles of sulfate and nitrate using WRF-Chem model (Weather Research and Forecasting Model coupled with online chemistry model in version 3.7.1). The results showed that sulfate was generally underestimated in WRF-Chem model at the upper level, while was in

relatively good agreement with observation at the surface. Possible reasons for the underestimations of sulfate are: (1) $SO_2$ precursors were underestimated at the upper levels (by about 45% to 77%, table S6), possibly due to the insufficient upward transport of $SO_2$ in the current model, especially in urban area where the urban canopy is low in resolution; (2) heterogeneous/multiphase formations of sulfate in droplets or aerosol water have not been considered enough in current model (Chen et al., 2016; Cheng et al. 2016).

The WRF-Chem model overestimated nitrate concentrations in both seasons. We put forward three potential explanations: (1) underestimation of $SO_4^{2-}$, which consumes less $NH_3$, facilitates $NH_3NO_4$ formation in the fine model at the high levels (Tuccella et al., 2012); (2) heterogeneous reaction of $HNO_3$ on coarse-mode sea salt aerosols, however, will reduce the formation of fine-mode nitrate (Chen et al, 2016). Therefore, sea salt emissions in current model should be evaluated especially over the coastal regions; (3) the cloud fraction and liquid water content may not be well simulated in the model. All these reasons would contribute to the highly varied nitrate concentrations at the upper levels. We added above discussions in the revision (section 3.4).

We also extended our discussions on the mass size distribution of OC and EC. We believe that the paper is now fit into the scope of the journal and can contribute important knowledge to aerosol sources and formation mechanisms in urban atmospheric boundary layer.

Please kindly find our following point-by-point response. The reviewer's comments are in black and the authors' responses are in blue. Any changes made in the revision are highlighted in red.

**Some specific comments:**
1) Page 2, lines 22-24 – When you say "highly uncertain", do you mean sources, formation mechanisms or both? I think the formation mechanisms, at least those that you focus on in this paper, are reasonably well known. There may be issues with the carbonaceous components that better fit 'highly uncertain', but you don't discuss those. Sources, because there are so many, may be highly uncertain, but you don't address those either.
*[A]*: We thank the reviewer for the comments. We have modified this sentence in the text (page 2, lines 23-27).
"Moreover, primary and secondary aerosols undergo chemical and physical processes, for example, transport, cloud processing, and removal from the atmosphere, leading to significantly spatial and temporal variations of the sources and formation mechanisms of atmospheric aerosols (Huang et al., 2014; Sun et al., 2015; Zhang et al., 2015; Liang et al., 2016)."

2) Page 3, lines 4 -6 - Direct emission is not a condensation process. Do you mean that the condensation mode results from condensation of gas-to-particle products condensing on direct emissions?
*[A]*: We thank the reviewer for the comments. We have modified this sentence in the

text (page 3, lines 8-10).

"The condensation submode originates from primary emissions and growth of smaller particles by coagulation and condensation, while droplet submode mainly results from cloud processing or coagulation of smaller particles (Seinfeld and Pandis, 2006)."

3) Page 3, lines 10 – 11 - There is evidence for this dating back to 1986 (e.g. Hoppel et al., JGR, 1985).

*[A]*: We thank the reviewer for pointing this out. We have added the reference in the text.

"Hoppel, W. A., Fitzgerald, J. W., and Larson, R. E.: Aerosol size distributions in air masses advecting off the east coast of the United States, J. Geophys. Res.-Atmos., 90, 2365-2379, 10.1029/JD090iD01p02365, 1985."

4) Page 3, line 14 – mechanical processes can also produce particles in the submicron size range.

*[A]*: We agree with the reviewer that mechanical processes can also produce particles in the submicron size range. We have rephrased this sentence.

"Coarse-mode particles are primarily produced by mechanical processes like sea spay, mineral particles and plant debris."

5) Page 4, line 3 – What are "very dynamic vertical profiles"?

*[A]*: We have modified "very dynamic vertical profiles" to "very dynamic changes in vertical concentration profiles"

6) Page 4, line 13 – remove "enough"

*[A]*: It has been removed from the text.

7) Page 4, line 15 – "aerosol pollution frequently occurs"

*[A]*: Changed.

8) Page 4, line 28 – page 5, line 3 - Instead, maybe say something like "Hence size-resolved vertical profiles of major components of PM can add to important knowledge of aerosol sources and formation mechanisms in the urban atmospheric boundary layer."

*[A]*: As suggested by the reviewer, we have modified this sentence.

Page 5, line 5-7: "Hence, size-resolved vertical concentration profiles of major components of PM can add to important knowledge of aerosol sources and formation mechanisms in the urban atmospheric boundary layer."

9) Page 5, lines 12-13 – I suggest something like "… were measured, and these observations are used to examine formation mechanisms and sources."

*[A]*: We thank the reviewer for the suggestion and we have modified the sentence in the manuscript.

10) Page 7, line 4 – reference date?

*[A]*: We have modified this sentence (Page 7, line 11):

"The principle of this model can be found in Draxler and Hess (1998)."

11) Page 7, line 24 – Please explain: e.g., do you mean things like insoluble components of dust?

*[A]*: We updated this sentence in the text (page 8, lines 2-4).

"The mismatch between the real-time $PM_{2.5}$ concentrations and the reconstructed $PM_{2.5}$ mass by combining the main components was likely due to sampling artefacts and lack of comprehensive offline analysis of $PM_{2.5}$ chemical components such as H and O associated with OC, geological minerals, and liquid water (Chow et al., 2015)."

12) Page 8, line 6 - Why not show $PM_1$ in the main text? It seems that $PM_1$ agrees much better with the sum of your ions, OC and EC, which would imply that much of the difference between your sum and $PM_{2.5}$ is due to water-insoluble material.

*[A]*: We thank the reviewer for pointing it out. The reasons for including $PM_{1.0}$ are: 1) $PM_{1.0}$ accounts for a large portion of $PM_{2.5}$ in southern China. Therefore, $PM_{1.0}$ agrees much better with the sum of ions, OC and EC. 2) $PM_{1.0}$ is not in the current air quality standard. In order to describe the pollution events, we focused on $PM_{2.5}$ and its components.

13) Page 9, lines 9-10 – They don't appear to be very dissimilar compared with the polluted day profiles.

*[A]*: The profiles in clean days are similar compared with the polluted day profiles. However, we notice that the concentrations of $PM_{2.5}$ and its components are much lower in the clean days than in the polluted days. In addition, high relative humidity and low clouds were observed in clean days. The main difference between clean days and polluted days in meteorological factors was temperature inversion as shown in Figure 7.

14) Paragraph starting "Figure 3" on pages 8-9 - This paragraph seems to have a take-away message that, in a relative sense, the carbonaceous components are more local and more of the inorganics (sulphate and nitrate) are transported from some distance. This might be a point to summarize.

*[A]*: A new sentence has been added on page 9, lines 14-16.

"The results suggest that the carbonaceous components are likably from local sources and the inorganics (sulfate and nitrate) are transported from some distances."

15) Page 9, line 14 – "cleanER"

*[A]*: Changed.

16) Page 9, lines 19-20 – "The reasons are unclear…"

*[A]*: Changed.

17) Page 10, line 1 – Some mention of RH is relevant here too.

*[A]*: New information has been added on page 10, lines 9-13:

"Ammonium nitrate is a temperature- and relative humidity-dependent compound, and low temperature and high RH facilitate the gas-to-particle partitioning (Wang et al., 2012; Bian et al., 2014). During the sampling periods, the average temperature in winter (13.5 ℃) was much lower than that in autumn (23.1 ℃) while the RH was vice versa, explaining higher concentration of $NH_4NO_3$ in winter than in autumn."

18) Page 10, lines 7-8 – If true, I suggest "… were the main components of the measured species, but the comparisons with $PM_1$ and $PM_{2.5}$ suggest these components were relatively small factors in the coarse particles."

*[A]*: We thank the reviewer for the suggestion and we have modified this sentence in the page 10, lines 18-19:

"Our results showed that OC, nitrate, crustal (e.g., $Ca^{2+}$, $Mg^{2+}$) and sea salt (e.g., $Na^+$ and $Cl^-$) were the major components of the measured species, but the comparisons with $PM_1$ and $PM_{2.5}$ suggest these components were relatively small in the coarse particles."

19) Page 10, lines 25-27 – Coagulation is important for smaller particles. Please clarify. Also, how likely are either in-cloud coalescence or multiple cloud cycles to impact your results. A reference for in-cloud coalescence is appropriate (e.g. Feingold and Kreidenweis).

*[A]*: Feingold et al. (1996) showed that collision-coalescence and aqueous phase chemistry have great impacts on the mass-mean size particles based on model simulations. We have modified this sentence and added one relevant reference on page 11, lines 10-12:

"It is generally recognized that coagulation is negligible at typical ambient particle number concentrations (Hinds, 1999). Therefore, in-cloud coalescence of droplets or aqueous phase chemistry is responsible for production of several sulfate modes (Feingold et al., 1996)."

**"3.4 Model simulation and implications**
Vertical concentration distributions of sulfate, nitrate and ammonium were further simulated by WRF-Chem model. The description and configurations of the model can be found in the Supplementary. Figures S13 and S14 show the simulated vertical concentration profiles of sulfate, nitrate and ammonium in autumn and winter and their comparisons with observation. Sulfate was generally underestimated in WRF-Chem model at the upper level, while was in relatively good agreement with observation at the surface. Possible reasons for the underestimations of sulfate are: (1) $SO_2$ precursors were underestimated at the upper levels (by about 45% to 77%, table S6), possibly due to the insufficient upward transport of $SO_2$ in the current model, especially in urban area where the urban canopy is low in resolution; (2) heterogeneous/multiphase formations of sulfate in droplets or aerosol water have not been considered enough in current model (Chen et al., 2016; Cheng et al. 2016). Nitrate was overestimated by WRF-Chem model. Here three reasons were put forward: (1) the underestimation of $SO_4^{2-}$ at the upper levels, which consumes less $NH_3$, facilitates the formation of $NH_3NO_4$ formation in the fine-mode (Tuccella et al., 2012); (2) heterogeneous reaction of $HNO_3$ on coarse-mode sea salt aerosols, however, will reduce the formation of fine-mode nitrate (Chen et al, 2016). Therefore, sea salt emissions in current model should be evaluated especially over the coastal regions; (3) the cloud fraction and liquid water content may not be well simulated in the model.

For ammonium, the simulated concentrations were overall consistent with the measured ones except for being slightly overestimated at ground level. The large discrepancies between observation and simulation on sulfate and nitrate suggested that physical and chemical mechanisms in current WRF-Chem model still need to be improved to better predict aerosol mass and composition. Based on our observation, in-cloud aqueous phase reactions and heterogeneous reactions should play important roles in sulfate and nitrate formation, which need to be refined in the model. Evaluation of WRF-Chem model incorporating the above-mentioned mechanisms is beyond the scope of this study and in-depth investigation needs to be done in future. Hence, more studies, such as long-term aerosols and high frequency micrometeorological measurements (Valiulis et al., 2002; Ceburnis et al., 2008; Ervens, 2015), are needed to identify the key aerosol sources and formation pathways, and to further improve the air quality models."

[Figure]

Figure S13. The vertical concentration profiles of sulfate, nitrate, and ammonium in $PM_{2.5}$ during autumn (The red solid lines are the average modeled concentrations and the shaded regions indicate the minimum and maximum values of the simulation; the average measurement data were in black with horizontal error bars).

[Figure]

Figure S14. The vertical concentration profiles of sulfate, nitrate, and ammonium in $PM_{2.5}$ during winter (The red solid lines are the average modeled concentrations and the shaded regions indicate the minimum and maximum values of the simulation; the average measurement data were in black with horizontal error bars).

**Supplementary materials**

**"2.1 Description and configurations of WRF–Chem model**

The model used in this study is the chemistry version of the WRF model (WRF–Chem v3.7.1). The WRF model is a mesoscale non-hydrostatic meteorological model that includes several options for physical parameterizations of the planetary boundary layer (PBL), cloud processes and land surface (Skamarock et al., 2008). The chemistry version is a version of WRF coupled with an "online" chemistry model, in which meteorological and chemical components of the model are predicted simultaneously (Grell et al., 2005; Fast et al., 2006). Emission, chemical formation and removal, transport and deposition are considered when WRF–Chem predicts the chemical components. In the numerical modeling, a triple-nested grid with 27/9/3 km resolution domain and 39 layers in vertical is set (Figure S12). The chemistry only run in domain d03 (3 km resolution) and MOZART boundary condition is used for the chemistry run. The simulation period is conducted from 0000 UTC 18 October to 0000 UTC 28 November 2015 as the autumn time period and 0000 UTC 25 December 2015 to 0000 UTC 28 January 2016 as the winter time period. The National Center for Environmental Protection (NCEP) $1° \times 1°$ FNL (Final operational global reanalysis) data are applied as the initial and boundary condition in WRF modeling. The physical and chemical parameterization configurations for WRF-Chem model can be found in Table S4. A region local anthropogenic emission inventory is used within the PRD region (Zheng et al., 2009), which is updated to the year of 2015. For other areas, the Multi-resolution Emission Inventory for China (MEIC, http://meicmodel.org/) is used in this study. MEGAN model version 2.1 is also used for providing the biogenic emissions (Guenther et al., 2006).

[Figure]

**Figure S12.** The simulation domain of WRF-Chem model

**Table S4.** Physical and chemical parameterization configurations for WRF-Chem model

| Process | Option |
| --- | --- |
| Microphysics scheme | Morrison (2 moments) |
| Cumulus scheme | Kain-Fristsch |
| Longwave radiation scheme | RRTM |
| Shortwave radiation scheme | Dudhia |
| Boundary-layer scheme | YSU |
| Land-surface scheme | unified Noah |
| Urban Surface scheme | UCM |
| Gas-phase mechanism scheme | CBM-Z |
| Photolysis scheme | Fast-J |
| Aerosol scheme | MOSAIC |

---

## Author Response (AR3)

Dear Professor Barbara Ervens,

Thank you for your letter and for the reviewer's comments concerning our manuscript entitled "*Vertical distribution of atmospheric particulate matter within the urban boundary layer in southern China: Size-segregated chemical composition and secondary formation through cloud processing and heterogeneous reactions*". Those comments are all valuable and very helpful for revising our paper, as well as the important guidance to our work. Accordingly, we made some changes in the revision to improve the manuscript.

We want to take this opportunity to explain the main motivation for this study. Though many measurements of atmospheric aerosols have been made at ground level, very few was focused on vertical measurements of size-resolved particulate matter within the urban boundary layer. We are fortunate to carry out long term measurements based on ~ 610 m tall Canton tower. It is innovation to reveal aerosol vertical distribution and formation in subtropical urban areas. It was found that cloud processing and heterogeneous reaction could be important formation pathways for the major aerosol compositions, contributing largely to air pollution in this urban region. In addition, we provided case studies to illustrate that cloud processing and heterogeneous reactions play important roles in the haze pollution in this area.

To broaden the scientific scope, we adopted the reviewer's suggestion to perform WRF-Chem simulation to see how well the model performance at reproducing the measurement results. Our results showed that model can simulate some components well such as ammonium and some components, such as sulfate, nitrate and OC, are poorly simulated. It indicated that the measurement data can be used to improve model in the future work. However, the comments were on the model improvement in this reviewer's suggestion. We agree that it is important for model development, but it is beyond the scope of this study. Hence, we still think this paper should focus primarily on measurement data analysis.

Once again, thank you very much for helping us with our manuscript. Major changes in the revision and point-to-point responds to the reviewer's comments are listed below. We hope that all the concerns raised by the reviewer shall be resolved.

Sincerely yours,

Shengzhen Zhou, PhD/Associate Professor
Xuemei Wang, PhD/Professor

**Response to referee**

The revised paper has improved, and it offers a useful look at the potentially more important processes in the evolution of the components of aerosol particles in this region. The authors enhanced the discussion of OC and EC, and they now include some comparisons with the results of WRF-Chem simulations. However, I find the comparisons and their discussion to be a little disappointing for the following reasons:
*[A]:* We would like to express our sincere appreciation for the reviewer's careful reading and invaluable comments to improve the paper. We tried our best to improve the manuscript and made some changes in the manuscript. Please kindly find our itemized responses to the specific comments below. The reviewer's comments are in black, and authors' responses are in blue. The changes in the manuscript are highlighted in red.

• The comparisons are not shown in the main text.
• There are no comparisons with OC, EC or NaCl.
• Other than to say that "it might not be well simulated", there is no attempt to assess the model's simulation of cloud with cloud observations (e.g. Figure S11), which is fundamental to their argument for cloud processing.
• Overall, the discussion seems hurried. It does not carefully consider all the major issues that may be associated with the comparison to their measurements and the use of the model to develop the conceptual picture they present in Figure 9.

The authors state that model evaluation "is beyond the scope of this study", yet they use the model results (e.g. Figure 7) to build their concept of the processes affecting haze formation in this region (Figure 9). This is not evaluation for the model. It should be evaluation to support your results, in other words, "does the model support their portrayal of the processes?".
*[A]:* Thank you for the comments. We have made lots of testing to WRF-Chem model, and some of the results are put in the text and supplementary.

"The performance statistics for meteorological elements such as pressure, air temperature, relative humility and wind speed of three vertical layers on the Canton Tower and chemical pollutants such as $PM_{2.5}$, $NO_x$, $O_3$ and $SO_2$ are shown in Table S5 and Table S6. Here, the statistical measures such as 5 Observation Mean, Simulation Mean, the Mean Bias (MB), the Normalized Mean Bias (NMB), the Normalized Mean Error (NME), the Mean Relative Bias (MRB), the Mean Relative Error (MRE), the Root Mean Squared Error (RMSE) and the correlation coefficient (CORR) are used for modeling validation.

**Table S5.** Comparison of Simulated Hourly Meteorological Elements with Observation Data

| Meteorology (Unit) | Height | number[a] | Mean Obs. | Sim. | MB | NMB[b] | NME[b] | RMSE | CORR |
|---|---|---|---|---|---|---|---|---|---|
| **Autumn** | | | | | | | | | |
| **PRES (hPa)** | GND | 925 | 1015.4 | 1013.7 | -1.7 | -0.2 | 0.2 | 1.8 | 0.98 |
| | 121m | 950 | 1002.0 | 1002.0 | 0.0 | 0.0 | 0.1 | 0.7 | 0.98 |
| | 454m | 952 | 959.9 | 959.2 | -0.7 | -0.1 | 0.1 | 1.3 | 0.94 |
| **TA (°C)** | GND | 952 | 24.8 | 23.7 | -1.2 | -4.7 | 6.8 | 2.1 | 0.92 |
| | 121m | 950 | 23.4 | 22.9 | -0.6 | -2.4 | 5.0 | 1.5 | 0.94 |
| | 454m | 952 | 20.6 | 20.8 | 0.2 | 0.9 | 3.9 | 1.1 | 0.95 |
| **RH (%)** | GND | 952 | 62.5 | 65.6 | 3.1 | 5.0 | 12.4 | 9.8 | 0.82 |
| | 121m | 950 | 64.9 | 67.1 | 2.2 | 3.4 | 11.2 | 9.1 | 0.85 |
| | 454m | 952 | 72.4 | 73.0 | 0.6 | 0.9 | 9.6 | 8.8 | 0.86 |
| **WS (m/s)** | GND | 952 | 0.7 | 2.3 | 1.6 | 227.0 | 234.8 | 2.1 | 0.62 |
| | 121m | 753 | 2.1 | 5.6 | 3.6 | 170.6 | 177.1 | 4.7 | 0.30 |
| | 454m | 936 | 4.1 | 6.6 | 2.5 | 60.2 | 74.6 | 4.0 | 0.55 |
| **Winter** | | | | | | | | | |
| **PRES (hPa)** | GND | 758 | 1021.2 | 1019.5 | -1.6 | -0.2 | 0.2 | 1.8 | 0.99 |
| | 121m | 770 | 1006.8 | 1007.8 | 1.1 | 0.1 | 0.1 | 1.3 | 0.99 |
| | 454m | 776 | 962.9 | 964.9 | 2.0 | 0.2 | 0.2 | 2.4 | 0.97 |
| **TA (°C)** | GND | 765 | 14.8 | 12.7 | -2.2 | -14.6 | 16.2 | 2.9 | 0.91 |
| | 121m | 525 | 14.7 | 13.5 | -1.1 | -7.7 | 10.5 | 1.8 | 0.94 |
| | 454m | 648 | 10.7 | 11.4 | 0.8 | 7.4 | 41.2 | 5.1 | 0.84 |
| **RH (%)** | GND | 765 | 67.0 | 68.4 | 1.4 | 2.1 | 14.4 | 11.9 | 0.84 |
| | 121m | 522 | 82.2 | 71.0 | -11.2 | -13.6 | 13.9 | 15.1 | 0.85 |
| | 454m | 245 | 73.9 | 54.2 | -19.7 | -26.7 | 28.1 | 25.1 | 0.85 |
| **WS (m/s)** | GND | 765 | 0.9 | 2.4 | 1.5 | 175.7 | 179.1 | 1.9 | 0.71 |
| | 121m | 526 | 2.0 | 6.4 | 4.4 | 215.5 | 220.6 | 5.7 | 0.23 |
| | 454m | 751 | 4.8 | 8.2 | 3.3 | 68.9 | 76.3 | 4.6 | 0.66 |

[a] the number of observed data

[b] the unit of NMB and NME is in %, other statistical variables are same as the meteorological element

**Table S6.** Comparison of Simulated Hourly Chemical Pollutants with Observation Data (unit: μg/m$^3$)

| Meteorology (Unit) | Height | number[a] | Mean Obs. | Sim. | MB | NMB[b] | NME[b] | RMSE | CORR |
|---|---|---|---|---|---|---|---|---|---|
| **Autumn** | | | | | | | | | |
| **PM$_{2.5}$** | GND | 958 | 43.6 | 42.6 | -1.0 | -2.3 | 59.9 | 35.9 | 0.24 |
| | 168m | 954 | 35.5 | 20.8 | -14.7 | -41.5 | 54.1 | 23.6 | 0.09 |
| | 488m | 947 | 27.5 | 13.0 | -14.5 | -52.8 | 59.9 | 19.9 | 0.22 |
| **NO$_x$** | GND | 949 | 101.7 | 90.7 | -10.9 | -10.7 | 62.0 | 93.4 | 0.41 |
| | 168m | 922 | 75.4 | 28.0 | -47.4 | -62.9 | 68.9 | 69.3 | 0.13 |

| | Height | | | | | | | | |
|---|---|---|---|---|---|---|---|---|---|
| | 488m | 950 | 27.1 | 7.7 | -19.3 | -71.5 | 79.7 | 28.9 | 0.24 |
| **O₃** | GND | 939 | 36.1 | 39.6 | 3.5 | 9.6 | 58.5 | 31.1 | 0.72 |
| | 168m | 947 | 58.3 | 72.0 | 13.8 | 23.6 | 62.9 | 46.6 | 0.64 |
| | 488m | 947 | 103.8 | 95.7 | -8.2 | -7.9 | 37.8 | 51.9 | 0.55 |
| **SO₂** | GND | 953 | 13.8 | 11.7 | -2.1 | -15.3 | 61.5 | 12.0 | 0.10 |
| | 168m | 949 | 17.8 | 5.1 | -12.7 | -71.4 | 72.4 | 16.7 | 0.03 |
| | 488m | 951 | 12.8 | 2.9 | -9.9 | -77.6 | 78.2 | 13.8 | 0.16 |
| **Winter** | | | | | | | | | |
| **PM₂.₅** | GND | 775 | 40.8 | 47.5 | 6.8 | 16.6 | 50.1 | 27.9 | 0.43 |
| | 168m | 756 | 33.0 | 28.7 | -4.3 | -12.9 | 42.0 | 20.5 | 0.35 |
| | 488m | 779 | 21.7 | 18.3 | -3.4 | -15.6 | 45.6 | 13.9 | 0.35 |
| **NOₓ** | GND | 755 | 115.4 | 167.1 | 51.7 | 44.8 | 83.4 | 147.7 | 0.45 |
| | 168m | 785 | 75.7 | 29.4 | -46.3 | -61.1 | 66.6 | 68.9 | 0.33 |
| | 488m | 705 | 25.6 | 5.7 | -20.0 | -77.9 | 85.5 | 29.5 | 0.28 |
| **O₃** | GND | 761 | 19.8 | 19.6 | -0.2 | -1.0 | 77.3 | 22.2 | 0.65 |
| | 168m | 786 | 26.6 | 61.3 | 34.7 | 130.1 | 137.5 | 43.3 | 0.49 |
| | 488m | 784 | 63.4 | 88.5 | 25.0 | 39.5 | 44.0 | 33.7 | 0.38 |
| **SO₂** | GND | 784 | 9.4 | 22.0 | 12.5 | 132.4 | 150.7 | 22.2 | 0.23 |
| | 168m | 786 | 11.8 | 7.8 | -4.1 | -34.5 | 48.2 | 8.8 | 0.05 |
| | 488m | 783 | 9.7 | 5.4 | -4.3 | -44.0 | 53.7 | 7.4 | 0.05 |

[a] the number of observed data

[b] the unit of NMB and NME is in %, other statistical variables are same as the meteorological variable

We have done our best to simulate the vertical distributions of major components of PM₂.₅, and found some potential problems to WRF-Chem model.

I suggest addressing that broader question by addressing the following more specific questions:

• **1.** Is it reasonable to expect that these data represent the grid box(es) that are being compared?

*[A]*: Thank you for the suggestions. In order to evaluate the general performance of the WFR-Chem as suggested, average data with uncertainties were plotted for the comparison between simulation and observation. We think this is a reasonable method for the comparison.

• **2.** How do the WRF-Chem simulations of OC, EC and NaCl compare to the observations? This will help to further identify model limitations.

*[A]*: We have compared concentration profiles of OC, EC and NaCl from the WRF-Chem simulations with those from our observations. The results show that, to some extent, the model still underestimate concentration of these species (Figure i). The WRF-Chem model setup and verification were given in the *supplementary.*

[Figure]

**Figure i.** The vertical concentration profiles of sulfate, nitrate, ammonium, OC, EC, sodium and chloride in $PM_{2.5}$ during (a) autumn and (b) winter field study (The red solid lines are the average modeled concentrations and the shaded regions indicate the

minimum and maximum values of the simulation; the average measurement data were in black with horizontal error bars).

• **3.** If WRF-Chem does not simulate sulphate in a reasonable way, then might it be because the temperature structure and winds of the lower levels are incorrect (as suggested by their statement that $SO_2$ was underestimated "possibly due to the insufficient upward transport of $SO_2$"), because $SO_2$ emissions are underestimated or because of a deficiency in the aqueous phase conversion of $SO_2$ to sulphate (e.g. representation of cloud; underestimation of oxidant concentrations; etc.)?

*[A]*: We totally agree that the underestimation of sulfate was possibly attributed to the underestimation of $SO_2$ precursor or a deficiency in the aqueous phase conversion of $SO_2$ to sulphate. These discussions were included in the last revision in response to the reviewer's insightful comments (page 17, lines 22-27):

"Sulfate was generally underestimated in WRF-Chem model at the upper level, while was in relatively good agreement with observation at ground level. Possible reasons for the underestimations of sulfate are: (1) $SO_2$ precursors were underestimated at the upper levels (by about 45% to 77%, Table S6), possibly due to the insufficient upward transport of $SO_2$ in the current model, especially in urban area where the urban canopy is low in resolution; (2) heterogeneous/multiphase formations of sulfate in droplets or aerosol water have not been fully considered in current model (Chen et al., 2016; Cheng et al. 2016)."

Our results present a unique vertical distribution dataset which should be useful for further model improvement. As suggested by the reviewer, many factors may contribute to inconsistency between simulation results and measurements, including meteorological parameters, cloud representation, underestimation of oxidant concentrations. We admitted that it is difficult to fully simulate the aerosol compositions using model such as WRF-Chem, especially in the vertical direction, which should still be an open question to the scientific community. This information was added in the manuscript (page 18, lines 7-16):

"The large discrepancies between observation and simulation on sulfate and nitrate suggested that physical and chemical mechanisms in current WRF-Chem model still need to be improved to better predict aerosol mass and composition. Based on our observation, in-cloud aqueous phase reactions and heterogeneous reactions should play important roles in sulfate and nitrate formation, which need to be refined in the model. Evaluation of WRF-Chem model incorporating the above-mentioned mechanisms is beyond the scope of this study and in-depth investigation needs to be done in future. Hence, more studies, such as long-term aerosols and high frequency micrometeorological measurements (Valiulis et al., 2002; Ceburnis et al., 2008; Ervens, 2015), are needed to identify the key aerosol sources and formation pathways, and to further improve the air quality models."

• **4.** Based on RH, it would seem that the model gets the clouds about right in the winter. Is that correct? What about autumn?

*[A]*: We detected and modeled low clouds both in autumn and in winter. However, the

density of the clouds in autumn was different from that in winter and much more clouds were observed in winter. We have provided some additional evidences in the manuscript: (1) The meteorological output from WRF-Chem simulations (Figure 8 in the manuscript); (2) The satellite images and ceilometer measurement (only in winter) (Figures S10 and S11 in the supplementary); (3) Records of in-situ observation (Figure ii).

[Figure]

**(a)**

[Figure]

**(b)**

**Figure 8.** Distribution of vertical wind (color scale, red: upward; blue: downward) and cloud fraction (black contour line) simulated by the WRF model during (a) autumn and (b) winter.

[Figure]

(a) November 18, 2015        (b) November 19, 2015

**Figure S10.** MODIS images show the cloud covers over the PRD region during the autumn pollution episode (https://earthdata.nasa.gov/earth-observation-data/near-real-time/rapid-response).

[Figure]

(a) January 02, 2016        (b) January 03, 2016

(c) Aerosol backscatter densities measured by ceilometer in Jan. 2 and Jan. 3, 2016.

**Figure S11.** Cloud cover from MODIS satellite remote sensing and cloud heights measured by ceilometer (Model CL-31, Vaisala Corp.) during the winter pollution episode.

[Figure]

**(a)**

**(b)**

**Figure ii.** Clouds in the (a) autumn haze episode; (b) winter haze episode (photos were taken at the 488 m platform of Canton tower at local time ~ 11:30 am).

• **5.** If upward transport is the problem, then is it valid to use the profiles in Figure 7 to develop the concept shown in Figure 9?

*[A]*: According to model simulations, airshed convection could induce upward transport, although it is still subjected to large uncertainties regarding the magnitude. Hence, we believe that the conceptual scheme in Figure 9 we propose is valid.

**Other comments:**

**1)** Page 1, line 21 – I find the statement starting with "Great progress has recently been made" needs to be referenced. I don't understand how great progress can be made at

ground level, without making it aloft. Ground level and the air above are connected. If you match things well at the ground, but not above, then it is not progress. I suggest the authors think more about their opening statement.

*[A]*: Thank you and we have rephased this sentence. Some references are in the introduction section.

"Many studies have recently been made on understanding the sources and formation mechanisms of atmospheric aerosols at ground level."

**2)** Page 2, line 4 - "droplet mode" is a less commonly used term. It has basis, but you need to discuss it first before using it. Also, it presumes the process that you are trying to establish. Use it in your discussion, but I suggest removing 'droplet' here and on line 7 as in my next comment.

*[A]*: We have deleted "droplet" in line 4.

**3)** Page 2, lines 7-10 – I suggest "Our results suggest that much of the sulfate and nitrate are formed from aqueous-phase reactions, and we attribute coarse mode nitrate formation at the measurement site to the heterogeneous reactions of gaseous nitric acid on existing sea-derived coarse particles in autumn.

*[A]*: We have changed the sentence (page 2, lines 7-9):

"Our results suggest that the majority of the sulfate and nitrate is formed from aqueous-phase reactions, and we attribute coarse mode nitrate formation at the measurement site to the heterogeneous reactions of gaseous nitric acid on existing sea-derived coarse particles in autumn."

**4)** Page 2, lines 10-11 - This sentence is repetitious. Reduce to "Case studies show that in combination with stagnant weather conditions, sulfate and nitrate from aqueous-phase and heterogeneous reactions contribute to haze formation during autumn and winter in the PRD region." Also, define 'PRD'.

*[A]*: We have changed accordingly.

"Case studies show that in combination with stagnant weather conditions, sulfate and nitrate from aqueous-phase and heterogeneous reactions contribute to haze formation during autumn and winter in the Pearl River Delta (PRD) region."

**5)** Page 3, lines 8-12 – I believe that 'droplet mode' resulted from "fog" studies. Often, there are significant differences between fog and cloud, and those differences are reflected in residual size distributions. It seems that you are knowledgeable about this, but you need to make it clear in this paragraph.

*[A]*: We agree that fog processing also leads to droplet mode residual particles. We have changed in the revision:

"The condensation submode particles originate from primary emissions and growth of smaller particles by coagulation and condensation, while droplet submode ones mainly result from cloud/fog processing or coagulation of smaller particles (Seinfeld and Pandis, 2006).

**6)** Page 4, line 15 – Change "the vertical size-resolved chemical composition" to "size-resolved chemical composition in the vertical"

*[A]*: It has been changed.

"However, measurements of size-resolved chemical composition in the vertical within the urban boundary layer are still lacking."

**7)** Page 4-5, lines 28-2 - "Additionally, more consistent evidences of aerosol formation through heterogeneous reactions are needed from field measurements, laboratory experiments and model simulations." Is awkward. Perhaps "Additionally, more studies of aerosol formation resulting from heterogeneous reactions are needed from field measurements, laboratory experiments and model simulations."

*[A]*: We have rephased in the revision.

"Additionally, more studies of aerosol formation resulting from heterogeneous reactions are needed from field measurements, laboratory experiments and model simulations."

**8)** Page 5, lines 17-20 – There is no mention of modelling here. Evaluation of the model is not one of your objectives, but the use of modelling in evaluating your observations is part of your work, and I think it should be mentioned.

*[A]*: We added the model evaluation in the objectives of this study (page 5, lines 20-22). (3) Evaluating the simulation performance of WRF-Chem model in the vertical based on the measurement data.

**9)** Page 8, line 25 – page 9, line 4 – I don't see the need for this classification. It adds unnecessary words, and you never use it beyond here. You only have 3 measurement altitudes, and you do not use "Type…" in any of the figures. Just start with ay something like, "We classify the ... components into those peaking at the ground, those peaking at 118 m and those peaking at 488 m", and modify the remainder of the paragraph accordingly. Also, I think these statistics should be in the main text, if you are going to discuss them here.

*[A]*: Thank you for the suggestion. We have changed as suggested. The present manuscript is very long. We think it's better to put the statistical tables in the supplementary.

**10)** Page 9, line 15 – "likely"; line 18 – "cleanER"

*[A]*: Changed.

**11)** Page 12, lines 15-18 – The proposition you refer to here was put forward decades ago. Re-write as "The coarse-mode nitrate was likely formed ... coarse particles (e.g. Anlauf ...)."

*[A]*: We agree and have changed in the manuscript (page 12, lines 17-20).

"The coarse-mode nitrate was likely formed the heterogeneous reactions of gaseous nitric acid with pre-existing sea- and soil-derived coarse particles (Anlauf et al., 2006; Harrison and Pio, 1983; Harrison and Kitto, 1990; Pakkanen, 1996; Wall et al., 1988;

Wu and Okada, 1994; Zhuang et al., 1999a)."

**12)** Page 12, line 20 - 6 - This equation isn't necessary. I suggest removing it and adding the references to the list beginning with Anlauf et al.
*[A]*: We have removed this equation and relevant sentences.

**13)** Page 12, lines 21-22 – This sentence repeats what is said four lines above. Remove it, and add the references to the "(Anlauf et al….)" list.
*[A]*: We have deleted the sentence and added the reference to the list as suggested.

**14)** Page 12-13, lines 28-1 - At best, "activation" is misleading in this context. Substantial water uptake by coarse NaCl particles can, and more likely occurs without true 'activation'. Replace with something like "Sea salt particles can grow by water uptake in fogs and clouds." I think this entire discussion could be better integrated and shortened, since the preference of HNO3 for NaCl over Calcium compounds that you mention is likely related to water solubility. I would make this point on line 25, and then refer to deliquesced particles, fog droplets and cloud droplets. As it stands now, it is as if the aqueous-phase processes are in addition to whatever drives your finding, stated on lines 22-24, and as further demonstrated by the repetitious statement on line 5 of page 13.
*[A]*: Thank the reviewer for the valuable suggestions. We rephased this part in the revision (Pages 12-13, lines 20-3).
"We found that coarse-mode $Na^+$, $Cl^-$, and $NO_3^-$ were at almost the same particle size, while $Ca^{2+}$ peaked at a particle size larger than $NO_3^-$ (Fig. S3). It is thus reasonable to conclude that coarse-mode $NO_3^-$ is probably associated with sea salt rather than $Ca^{2+}$, which is consistent with the previous work in Hong Kong (Zhuang et al., 1999b). Sea salt particles can grow by water uptake in fogs and clouds. A previous study showed that a substantial amount of nitrates forms when $HNO_3$ reacts with deliquesced sea-salt as compared to the dry NaCl particles (Brink, 1998). Hence, we speculated that nitrates were formed from the reactive uptake of $HNO_3$ in the deliquesced sea salt droplets rather than dry particles in Guangzhou. The back-trajectory cluster analysis showed that the sampled air masses were predominantly from the South China Sea and moved toward Guangzhou in autumn (Fig. S4), bringing high concentrations of sea salt particles available for heterogeneous reactions. Moreover, high relative humidity, fog, and low clouds which were observed during the observation, could facilitate the heterogeneous formation of coarse-mode nitrates."

**15)** Page 14, line 2 – "were slightly higher"
*[A]*: We have changed the phrase.

**16)** Page 14, line 11 – "demonstrating more aged aerosol" - Are you saying this simply because the aerosol was sampled at a higher elevation?
*[A]*: Our results showed that higher chloride depletion in the higher elevation, suggesting the aerosols were more aged.

**17)** Page 14, lines 12-15 – Might some of the winter salt particles be attributed to dry salt lakes in the interior of China?

*[A]*: It is possible that salt particles can be originated from remote dry salt lakes, such as the lakes in Qinghai province in northwestern China. The salt particles from lakes may be carried along with the dust storms, which usually occur in Spring. We think dry salt lakes were not the sources of winter salt particles in our measurement.

**18)** Page 14, line 18 – "smallER"

*[A]*: It has been changed.

**19)** Page 14, lines 21-22 – But the profiles of SO4=and OC differ. Secondary contributions to SO4= appear to be more elevated than OC, at least in autumn.

*[A]*: We agree that $SO_4^{2-}$ could be more elevated at the higher levels due to the aqueous phase reaction.

Organic aerosols can originate from direct emission (primary organic carbon, POC) or atmospheric secondary formation (secondary organic carbon, SOC). The ground OC sources would impact the vertical distributions of OC. However, sulfate is produced mainly from atmospheric reactions. That could be the reason why $SO_4^{2-}$ and OC had different vertical profiles.

**20)** Page 14, line 24 - The ratio is determined by 2 quantities. A change in the ratio doesn't necessarily result from a change in just one of those quantities. Maybe the ratio at the source differs, or maybe precipitation near the source remove some of the particles, including EC, after which the sulphate, nitrate and OC are replenished by secondary reactions. The statement, "indicating the presence of secondary organic aerosols", needs more justification.

*[A]*: Besides the OC/EC ratios, we estimated the secondary organic carbon (SOC) concentrations using the EC tracer method in the revised manuscript. We found that SOC accounted for a large fraction of OC (Figure S6). We added this information in the text, and the EC-tracer method was included in the supplementary.

"The high OC/EC ratios were found in particles with sizes larger than 0.25 μm, especially for droplet mode particles, indicating the enhancement of the secondary formation of OC in this mode. We further evaluated the contributions of secondary organic carbon (SOC) to OC using EC-tracer method (Castro et al., 1999; Zhou et al. 2014). The results showed that SOC accounted for a large fraction of OC in our study (Figure S6)."

[Figure]

**Figure S6.** Average OC/EC ratios and percentages of SOC in OC at different sizes during autumn and winter.

**3. Estimation of secondary organic carbon concentration**

It is difficult to separate primary organic carbon (POC) from secondary organic carbon (SOC). Currently, T no simple, direct analytical technique is available for the separation. Here we applied an indirect method (the minimum OC/EC ratio method) to estimate the secondary organic carbon ($OC_{sec}$) formation with the following equation (Cao et al., 2004; Castro et al., 1999; Zhou et al., 2014):

$$OC_{sec} = OC_{tot} - EC \times (OC/EC)_{min}$$

where $OC_{sec}$ is secondary organic carbon, $OC_{tot}$ is total organic carbon, EC is elemental carbon, and (OC/EC) min is the minimum OC/EC ratio. The minimum ratio of OC/EC may be affected by many factors such as meteorological conditions, the variation of the emission source, and the transport of aged aerosols. Thus, SOC concentration derived by the EC tracer method could be underestimated in this study. In order to reduce this uncertainty, the $(OC/EC)_{min}$ is defined as the minimum OC/EC ratio of each size interval at each height in autumn and winter respectively.

References:

Cao, J. J., Lee, S. C., Ho, K. F., Zou, S. C., Fung, K., Li, Y., Watson, J. G., and Chow, J. C.: Spatial and seasonal variations of 25 atmospheric organic carbon and elemental carbon in Pearl River Delta Region, China, Atmos. Environ., 38, 4447-4456, https://doi.org/10.1016/j.atmosenv.2004.05.016, 2004.

Castro, L.M., Pio, C.A., Harrison, R.M., Smith, D.J.T., 1999. Carbonaceous aerosol in urban and rural European atmospheres: estimation of secondary organic carbon concentrations. Atmos. Environ. 33, 2771e2781. http://dx.doi.org/10.1016/s1352-2310(98)00331-8.

Zhou, S. Z., Wang, T., Wang, Z., Li, W. J., Xu, Z., Wang, X. F., Yuan, C., Poon, C. N., Louie, Peter K.K., Luk, Connie W.Y., and Wang, W. X: Photochemical evolution of organic aerosols observed in urban plumes from Hong Kong and the Pearl River Delta of China, Atmos. Environ., 88, 219–229, http://dx.doi.org/10.1016/j.atmosenv.2014.01.032, 2014.

**21)** Page 15, lines 24-25 – What about radiative cooling?

*[A]*: Radiative cooling usually occurs under the circumstances of cloud free. As showed in *Figure S10 and S11*, there were many clouds during these periods. And, wind directions were changed from south to north (*Figure S8*). Therefore, we thought that temperature inversion in our study should be caused by the convergence of two different air streams.

**22)** Page 16, line 1 – Should be Fig. 7a?

*[A]*: It has been changed.

**23)** Page 16, lines 14-15 – "Our results suggest that aqueous-phase and heterogeneous reactions contributed significantly to the sulfate and nitrate in the PRD region during this episode.

*[A]*: The sentence has been changed to (page 16, lines 12-13):

"Our results suggest that aqueous-phase and heterogeneous reactions contributed significantly to the sulfate and nitrate in the PRD region during this episode."

**24)** Page 16, line 2 – As in my first review, this is NOT "strong convection". I suggest "Low-level cloud was observed during this period, associated with weak convection simulated by WRF."

*[A]*: We have changed in the revised manuscript (page 16, lines 20-21).

"Low-level cloud was observed during this period, associated with weak convection simulated by WRF."

**25)** Page 17, line 3 – What is meant by "aggravated"?

*[A]*: We changed to "facilitated".

**26)** Page 17, discussion at end of Section 3.3 – There is no discussion of precipitation in the paper. Was there precipitation? If so, might it have played a role?

*[A]*: There was no rain during two aerosols pollution events we discussed.

Page 15, lines 13-14: "There was no rain during the pollution episodes."

**27)** Page 17, line 23 – Figures S13 and S14 should be in the main text.

*[A]*: We moved these figures to the main text as Figure 10.

[Figure]

Figure 10. The vertical concentration profiles of sulfate, nitrate, and ammonium in PM$_{2.5}$ during (a) autumn and (b) winter (The red solid lines are the average modeled concentrations and the shaded regions indicate the minimum and maximum values of the simulation; the average measurement data were in black with horizontal error bars).

**28)** Page 18, line 21 – "Vertical characteristics and potential formation processes of…"
*[A]*: This sentence has been changed accordingly.
"Vertical characteristics and potential formation processes of size-resolved aerosols were studied during autumn and winter seasons utilizing the 610 m Canton Tower in Guangzhou."

---

## Author Response (AR4)

Dear Professor Barbara Ervens,

Thank you for your valuable comments and suggestion. We would like to publish this paper as a Measurement Report. Please find below the response to the minor/technical comments. Finally, we sincerely thank you for handling this manuscript, and we have also learned a lot from your expertise.

Best regards,

Shengzhen Zhou, PhD/Associate Professor
Xuemei Wang, PhD/Professor

**Response to minor/technical comments**

**1.** p. 1, l. 25: replace 'were' by 'was'
*[A]:* Changed.

**2.** p. 3, l. 20: replace 'reaction mechanisms' by 'reactions'
*[A]:* Changed.

**3.** p. 5, l. 1: This sentence seems to partially repeat the text from the previous one: "quantification of the aerosol formation through heterogeneous reactions under real atmospheric conditions" and "more studies of aerosol formation resulting from heterogeneous reactions are needed from field measurements". Please reword or shorten.
*[A]:* We have rephased these sentences.
"To improve the understanding of haze formation, models will require updated kinetic and mechanistic data of multiphase chemistry, and quantification of the aerosol formation through heterogeneous reactions are needed from field measurements, laboratory experiments and model simulations under real atmospheric conditions."

**4.** p. 9, l. 22: replace 'likably' by 'likely'

*[A]:* Changed.

**5.** p. 12, l. 17: add 'by' (formed by the heterogeneous reactions…)
*[A]:* Changed.

**6.** p. 14, l. 18: replace 'much' by 'very'
*[A]:* Changed.

**7.** p. 14, l. 20: 'Jaffrezo' misspelled
*[A]:* Changed.

**8.** p. 16, l. 21: Did you use WRF or WRF-Chem for these simulations?
*[A]:* We applied WRF to simulate the distribution of vertical wind and cloud fraction. Vertical concentration distributions of sulfate, nitrate and ammonium were further simulated by WRF-Chem as suggested by the referee.

**9.** p. 16, l. 22: add 'on' (than on event days)
*[A]:* Changed.

**10.** p. 17, l. 20: 'Figure 10 …'
*[A]:* Changed to "Figure 10"

**11.** p. 17, l. 22 : add 'it' (while it was in ….)
*[A]:* Changed.